# BOPO: Neural Combinatorial Optimization via Best-anchored and Objective-guided Preference Optimization

**Zijun Liao** [* 1]  **Jinbiao Chen** [* 1]  **Debing Wang** [1]  **Zizhen Zhang** [† 1]  **Jiahai Wang** [1]

## Abstract

Neural Combinatorial Optimization (NCO) has emerged as a promising approach for NP-hard problems. However, prevailing RL-based methods suffer from low sample efficiency due to sparse rewards and underused solutions. We propose *Best-anchored and Objective-guided Preference Optimization (BOPO)*, a training paradigm that leverages solution preferences via objective values. It introduces: (1) a best-anchored preference pair construction for better explore and exploit solutions, and (2) an objective-guided pairwise loss function that adaptively scales gradients via objective differences, removing reliance on reward models or reference policies. Experiments on Job-shop Scheduling Problem (JSP), Traveling Salesman Problem (TSP), and Flexible Job-shop Scheduling Problem (FJSP) show BOPO outperforms state-of-the-art neural methods, reducing optimality gaps impressively with efficient inference. BOPO is architecture-agnostic, enabling seamless integration with existing NCO models, and establishes preference optimization as a principled framework for combinatorial optimization.

## 1. Introduction

Combinatorial optimization problems (COPs), such as scheduling (Zhang et al., 2019; Xiong et al., 2022) and routing problems (Vidal et al., 2020; Berghman et al., 2023), are widely applied in real-world scenarios and have attracted significant research attention. Most COPs are NP-hard, making them challenging to find optimal solutions. Exact methods, such as branch-and-bound algorithms, require exponential computation time as the problem size increases. Consequently, heuristic methods have proven effective in obtaining high-quality solutions within reasonable time over the past decades. Nevertheless, these methods still heavily rely on expert knowledge and extensive iterative search.

In the emerging field of neural combinatorial optimization (NCO), deep neural models are employed to automatically learn heuristics from training data, enabling the rapid construction of high-quality solutions in an end-to-end fashion (Mazyavkina et al., 2021; Bengio et al., 2021; Yan et al., 2022; Kayhan & Yildiz, 2023; Zhang et al., 2023; Garmendia et al., 2024). Early research (Vinyals et al., 2015) adopted supervised learning (SL) to train the deep models, which required (near-) optimal solutions produced by expensive specialized solvers as labels. Different from SL, reinforcement learning (RL), which does not require labels, has emerged as the mainstream training paradigm for NCO (Kool et al., 2019; Kwon et al., 2020; Zhang et al., 2020; Ho et al., 2024). However, RL encounters challenges such as sparse rewards and low sample efficiency (Kim et al., 2024). Recently, self-labeling learning (SLL) (Corsini et al., 2024; Luo et al., 2025) was proposed to partially address these issues by sampling multiple solutions and treating the best one among them as a pseudo-label for model training. Nevertheless, SLL still faces the challenge of low sample efficiency, as all sampled solutions except the optimal one are discarded during training.

To improve sample efficiency in NCO training, we leverage multiple sampled solutions rather than focusing solely on the optimal one by introducing preference optimization (Rafailov et al., 2023; Meng et al., 2024). To this end, we propose *Best-anchored and Objective-guided Preference Optimization (BOPO)*, which leverages the natural preference relation among solutions according to their objective values. Our approach builds upon two fundamental observations: (1) NCO models (typically generative models) can generate multiple distinct solutions for a given problem instance, and (2) the objective value of a COP solution can be computed with a low cost. As shown in Figure 1, our BOPO comprises two essential components: constructing multiple preference pairs from sampled solutions and building a preference optimization loss. As a new training paradigm for neural combinatorial optimization, BOPO avoids expensive labels and formulation of Markov Decision Process, making

---

[*]Equal contribution  [1]School of Computer Science and Engineering, Sun Yat-sen University, China. Correspondence to: Zizhen Zhang <zhangzzh7@mail.sysu.edu.cn>.

*Proceedings of the 42nd International Conference on Machine Learning*, Vancouver, Canada. PMLR 267, 2025. Copyright 2025 by the author(s).

it particularly well-suited for addressing various types of COPs.

In summary, our contributions are as follows:

- We propose BOPO, a novel training paradigm using preference optimization for neural combinatorial optimization, which enhances sample efficiency compared with the mainstream RL and recent SLL paradigms.

- As the first key component of BOPO, we design a best-anchored preference pair construction method for COPs to better explore and exploit solutions.

- As the second key component, we tailor a novel objective-guided preference optimization loss that incorporates preferences quantified by objective values of COPs.

- Experimental results on three classic problems, namely the Job-shop Scheduling Problem (JSP), Traveling Salesman Problem (TSP), and Flexible Job-shop Scheduling Problem (FJSP), demonstrate that BOPO outperforms state-of-the-art methods.

## 2. Related Works

**Supervised Learning (SL) for NCO.** SL methods utilize optimal solutions as labels to train neural models with cross-entropy loss for solving COPs, such as TSP (Vinyals et al., 2015; Milan et al., 2017) and JSP (Ingimundardottir & Runarsson, 2018). Data augmentation techniques have been exploited to enhance the performance of SL methods for routing problems (Luo et al., 2023; Yao et al., 2024). Additionally, diffusion-based SL approaches (Sun & Yang, 2023; Li et al., 2023; 2024) learn to generate heatmaps for TSP. The primary limitation of SL, however, lies in the high computational cost of obtaining optimal solutions as labels, which restricts its practical applications.

**Reinforcement Learning (RL) & Self-Labeling Learning (SLL) for NCO.** Label-free RL is currently the mainstream training paradigm in neural combinatorial optimization. Attention Model (Kool et al., 2019), which combines RL with Transformer (Vaswani, 2017), marks a milestone in solving routing problems. The policy optimization with multiple optima (POMO) (Kwon et al., 2020) introduces a shared baseline leveraging solution symmetry. Given its competitive performance and practicality, POMO has established itself as a prominent training algorithm for routing problems and has inspired numerous advancements (Grinsztajn et al., 2023; Chalumeau et al., 2023; Drakulic et al., 2023; Chen et al., 2023a;b; Xiao et al., 2024b; Goh et al., 2024; Fang et al., 2024; Zhou et al., 2024a;b; Bi et al., 2024; Zheng et al., 2024; Wang et al., 2024; Chen et al., 2025a;b). Additionally, RL has been widely adopted for scheduling problems,

including JSP (Zhang et al., 2020; Park et al., 2021a;b; Jeon et al., 2023; Tassel et al., 2023; Iklassov et al., 2023; Ho et al., 2024) and FJSP (Song et al., 2023; Yuan et al., 2024). Different from the above RL-based constructive methods, RL is also employed in improvement methods for both routing (Ma et al., 2021; Wu et al., 2022; Ma et al., 2023; Kong et al., 2024) and scheduling problems (Falkner et al., 2022; Zhang et al., 2024b). Recently, SLL (Luo et al., 2025; Corsini et al., 2024) utilizes the local optimal solution during training as a pseudo-label to train an end-to-end model using cross-entropy loss, where self-labeling improvement method (SLIM) (Corsini et al., 2024) has achieved state-of-the-art performance on JSP.

**Preference Optimization.** Preference optimization has been widely adopted to align large language models (LLMs) with human preferences. One of the most well-known techniques is reinforcement learning with human feedback (RLHF) (Stiennon et al., 2020; Ouyang et al., 2022), which trains a reward model using ranking learning and then aligns LLM through RL. Direct preference optimization (DPO) (Rafailov et al., 2023) offers an efficient alternative by skipping the reward model training phase and directly optimizing LLMs using preference pairs. Building upon DPO, subsequent studies explore comparing more samples (Dong et al., 2023; Song et al., 2024) and designing more concise loss functions (Xu et al., 2023; Meng et al., 2024). Among them, simple preference optimization (SimPO) (Meng et al., 2024) has gained popularity due to its simplicity and effectiveness. Recent advancements in preference optimization are summarized in a comprehensive survey (Xiao et al., 2024a). Inspired by preference optimization, we propose best-anchored and objective-guided preference optimization (BOPO). A concurrent work (Pan et al., 2025) also applies preference optimization to COPs. However, fundamentally different from Pan et al. (2025), BOPO develops a best-anchored preference pair construction method and a novel objective-guided preference optimization loss specially designed for COPs.

## 3. Preliminaries

### 3.1. Neural Combinatorial Optimization (NCO)

COP aims to find a solution $\boldsymbol{y}$ that minimizes (or maximizes) the objective function $g(\boldsymbol{y})$. In the NCO domain, neural constructive methods sequentially construct a solution $\boldsymbol{y}$ in an end-to-end manner for a COP instance $\boldsymbol{x}$. Specifically, at step $t \in \{1, \cdots, |\boldsymbol{y}|\}$, a feasible action $y_t$ is selected based on the partial solution $\boldsymbol{y}_{<t} = (y_1, \cdots, y_{t-1})$ with constraints enforced through masking. A model with parameter $\boldsymbol{\theta}$ outputs the policy $\pi_{\boldsymbol{\theta}}(\boldsymbol{y}|\boldsymbol{x}) = \prod_{t=1}^{|\boldsymbol{y}|} \pi_{\boldsymbol{\theta}}(y_t|\boldsymbol{y}_{<t}, \boldsymbol{x})$ of solution $\boldsymbol{y}$. Solutions can be obtained via multiple search strategies based on policy $\pi_{\boldsymbol{\theta}}(\boldsymbol{y}|\boldsymbol{x})$, including greedy and sampling rollouts.

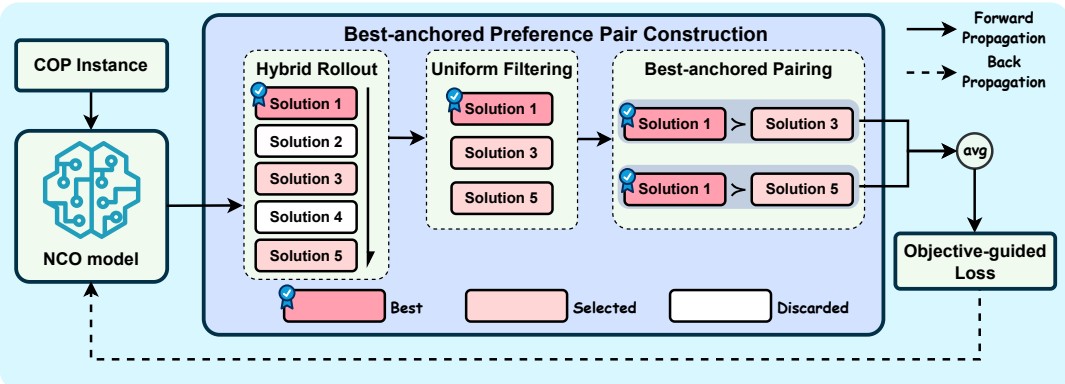

*Figure 1.* The pipeline of best-anchored and objective-guided preference optimization (BOPO).

Typical training paradigms include SL, RL, and SLL. SL utilizes the (near-) optimal solution $\boldsymbol{y}^*$ as a label to train model $\boldsymbol{\theta}$ using cross-entropy loss $\mathcal{L}(\pi_{\boldsymbol{\theta}}|\boldsymbol{y}^*, \boldsymbol{x}) = -\log \pi_{\boldsymbol{\theta}}(\boldsymbol{y}^*|\boldsymbol{x})$. In RL, the REINFORCE loss (Williams, 1992) $\mathcal{L}(\pi_{\boldsymbol{\theta}}|\boldsymbol{y}, \boldsymbol{x}) = -(g(\boldsymbol{y}) - b(\boldsymbol{x}))\log \pi_{\boldsymbol{\theta}}(\boldsymbol{y}|\boldsymbol{x})$ with a baseline $b(\boldsymbol{x})$ is commonly used for routing problems, while the proximal policy optimization (PPO) (Schulman et al., 2017) loss $\mathcal{L}(\pi_{\boldsymbol{\theta}}|\boldsymbol{y}, \boldsymbol{x}) = -\sum_t \min\{r_t \hat{A}_t, \text{clip}(r_t, 1 - \epsilon, 1 + \epsilon)\hat{A}_t\}$ is predominantly used for scheduling problems, where $r_t = \frac{\pi_{\boldsymbol{\theta}}(y_t|\boldsymbol{y}_{<t}, \boldsymbol{x})}{\pi_{\boldsymbol{\theta}'}(y_t|\boldsymbol{y}_{<t}, \boldsymbol{x})}$ denotes the ratio of current policy $\boldsymbol{\theta}$ and old policy $\boldsymbol{\theta}'$, $\hat{A}_t$ represents the advantage estimate, and $\epsilon$ is a hyperparameter. SLL, a recent paradigm, selects the best among solutions sampled from current policy $\boldsymbol{\theta}$ as a pseudo-label and applies cross-entropy loss.

### 3.2. Solution Construction for Classic COPs

The job-shop scheduling problem (JSP) entails allocating a set of $n$ jobs across $m$ machines with shape $(n \times m)$, wherein each job must be performed on the machines in a predefined sequence. The instance $\boldsymbol{x}$ comprises the processing time of operations and the corresponding required machines. Action $y_t$ is defined as an operation that assigns a job to the earliest available time slot on the corresponding machine and updates the job's progress. The goal is to determine the job processing order on each machine to minimize the maximum completion time, known as the *makespan*. This construction involves $|\boldsymbol{y}| = nm$ steps.

For the traveling salesman problem (TSP), the instance $\boldsymbol{x}$ is composed of $n$ nodes with 2-dimensional coordinates. The objective is to find a tour that passes through all nodes with minimal total distance. To construct a TSP solution, an unvisited node $y_t$ at step $t$ is selected to be added to the current partial tour. This process requires $|\boldsymbol{y}| = n$ steps.

The flexible job-shop scheduling problem (FJSP) extends JSP by considering that each operation can be processed on

multiple candidate machines with shape $(n \times m \times k)$, where $k$ denotes the maximum number of operations in all jobs. To construct a FJSP solution, action $y_t$ at step $t$ represents a joint selection of an operation and one of its available machines. This construction requires $|\boldsymbol{y}| = nk$ steps.

The definitions of above COPs and details of their features are provided in Appendix A.

## 4. Methodology

### 4.1. Best-anchored and Objective-guided Preference Optimization (BOPO)

To improve sample efficiency, we propose a novel training paradigm, namely BOPO. Distinct from the RL and SLL training paradigms, BOPO exploits the preference relations among generated solutions according to their objective values. Specifically, for a COP with the minimization objective, the *explicit preference* $f^*(\boldsymbol{y}, \boldsymbol{x})$ for instance $\boldsymbol{x}$ and solution $\boldsymbol{y}$ is defined as the negative of the objective function:

$$f^*(\boldsymbol{y}, \boldsymbol{x}) = -g(\boldsymbol{y}). \tag{1}$$

A *preference pair*, denoted as a triplet $(\boldsymbol{x}, \boldsymbol{y}_w, \boldsymbol{y}_l)$, consists of an instance $\boldsymbol{x}$ and two solutions $\boldsymbol{y}_w$ and $\boldsymbol{y}_l$ satisfying $\boldsymbol{y}_w \succ \boldsymbol{y}_l \triangleq f^*(\boldsymbol{y}_w, \boldsymbol{x}) > f^*(\boldsymbol{y}_l, \boldsymbol{x})$.

BOPO employs a preference optimization loss based on such preference pairs to train the neural model parameterized as $\boldsymbol{\theta}$. As the two critical components of BOPO, we develop a best-anchored preference pair construction method and derive a novel objective-guided preference optimization loss function specialized for COPs.

### 4.2. Best-anchored Preference Pair Construction

The construction of preference pairs consists of three steps: (1) **Hybrid Rollout** generates diverse solutions via sampling rollout and a high-quality one via greedy rollout. (2) **Uni-**

**form Filtering** selects representative ones from the obtained solutions for efficient pairing. (3) **Best-anchored Pairing** constructs preference pairs to enhance model learning.

**Hybrid Rollout.** Both diverse and high-quality solutions play vital roles in model learning. Sampling from policy $\pi_{\boldsymbol{\theta}}(\boldsymbol{y}, \boldsymbol{x})$ generates diverse solutions, occasionally surpassing the greedy rollout solution. However, most sampled solutions are inferior to the greedy one. To leverage their complementary strengths, we propose a hybrid rollout strategy combining both approaches. This strategy generates $B$ solutions, including $B - 1$ from sampling and one from greedy rollout. It ensures coverage of both exploratory and exploitative solutions.

**Uniform Filtering.** Constructing preference pairs using all $B$ solutions would produce a combination of $\binom{B}{2}$ pairs, resulting in high computational cost and many low-quality pairs. Instead, we employ uniform filtering to select solutions to maximize representational diversity. Specifically, we select $K$ solutions $\mathcal{C} = \{\boldsymbol{y}_1 \succ \cdots \succ \boldsymbol{y}_K\}$ uniformly from $B$ sorted solutions $\mathcal{S} = \{\boldsymbol{y}'_1 \succ \cdots \succ \boldsymbol{y}'_B\}$, i.e., $\boldsymbol{y}_k = \boldsymbol{y}'_{\lfloor B/K \rfloor \cdot (k-1)+1}, \forall k \in \{1, \cdots, K\}$. This avoids overfitting to clusters of similar solutions.

**Best-anchored Pairing.** Since a COP solely focuses on finding the optimal solution, we anchor pairs to the best solution to prioritize learning from high-quality examples. For $K$ solutions $\{\boldsymbol{y}_1 \succ \cdots \succ \boldsymbol{y}_K\}$, we create $K - 1$ preference pairs, each combining the best solution with a suboptimal one, i.e., $\mathcal{P} = \{(\boldsymbol{x}, \boldsymbol{y}_1, \boldsymbol{y}_k) | k \in \{2, \cdots, K\}\}$. This design encourages learning from the optimal solution while discouraging learning from various suboptimal ones, being more efficient than using all $\binom{K}{2}$ possible pairs.

### 4.3. Objective-guided Preference Optimization Loss

After obtaining preference pairs, we formulate the loss function of BOPO by incorporating a preference-based scaling factor derived from the objective values of COP solutions.

**Objective-guided Preference Optimization Loss.** For policy $\pi_{\boldsymbol{\theta}}(\boldsymbol{y}, \boldsymbol{x})$ used to construct solution $\boldsymbol{y}$, its *implicit preference* $f_{\boldsymbol{\theta}}(\boldsymbol{y}, \boldsymbol{x})$ is defined as the average log-likelihood:

$$f_{\boldsymbol{\theta}}(\boldsymbol{y}, \boldsymbol{x}) = \frac{1}{|\boldsymbol{y}|} \log \pi_{\boldsymbol{\theta}}(\boldsymbol{y}|\boldsymbol{x}) = \frac{1}{|\boldsymbol{y}|} \sum_{t=1}^{|\boldsymbol{y}|} \log \pi_{\boldsymbol{\theta}}(y_t|\boldsymbol{y}_{<t}, \boldsymbol{x}). \tag{2}$$

For a preference pair $(\boldsymbol{x}, \boldsymbol{y}_w, \boldsymbol{y}_l)$, the *preference distribution* $p_{\boldsymbol{\theta}}(\boldsymbol{y}_w \succ \boldsymbol{y}_l | \boldsymbol{x})$ is modeled using the Bradley-Terry ranking objective (Bradley & Terry, 1952) and implicit preferences:

$$\begin{aligned} p_{\boldsymbol{\theta}}(\boldsymbol{y}_w \succ \boldsymbol{y}_l | \boldsymbol{x}) = \\ \sigma(\beta(\boldsymbol{x}, \boldsymbol{y}_w, \boldsymbol{y}_l)(f_{\boldsymbol{\theta}}(\boldsymbol{y}_w, \boldsymbol{x}) - f_{\boldsymbol{\theta}}(\boldsymbol{y}_l, \boldsymbol{x}))), \end{aligned} \tag{3}$$

where $\sigma(\cdot)$ is the sigmoid function. $\beta(\boldsymbol{x}, \boldsymbol{y}_w, \boldsymbol{y}_l) = f^*(\boldsymbol{y}_l, \boldsymbol{x})/f^*(\boldsymbol{y}_w, \boldsymbol{x})$ is a preference-based adaptive scal-

---

**Algorithm 1** BOPO Training

1: **Input:** Dataset $\mathcal{X}$, number of epochs $E$, number of training steps $T$, batch size $D$, number of obtained solutions $B$, number of filtered solutions $K$, and learning rate $\eta$
2: Initialize model parameter $\boldsymbol{\theta}$
3: **for** $epoch = 1$ **to** $E$ **do**
4:     **for** $step = 1$ **to** $T$ **do**
5:         $\boldsymbol{x}_i \leftarrow \text{SAMPLEINSTANCE}(\mathcal{X}) \, \forall i \in \{1, \ldots, D\}$
6:         $\mathcal{S}_i \leftarrow \text{HYBRIDROLLOUT}(\boldsymbol{x}_i, B) \, \forall i \in \{1, \ldots, D\}$
7:         $\mathcal{C}_i \leftarrow \text{UNIFORMFILTERING}(\mathcal{S}_i, K) \, \forall i \in \{1, \ldots, D\}$
8:         $\mathcal{P}_i \leftarrow \text{BEST-ANCHOREDPAIRING}(\mathcal{C}_i) \, \forall i \in \{1, \ldots, D\}$
9:         Compute $\mathcal{L}_{BOPO}(\pi_{\boldsymbol{\theta}}, \boldsymbol{x}, \boldsymbol{y}_w, \boldsymbol{y}_l)$ using Equation (4)
10:        $\mathcal{L}(\boldsymbol{\theta}) \leftarrow \frac{1}{D} \sum_{i=1}^{D} \frac{1}{|\mathcal{P}_i|} \sum_{(\boldsymbol{x}, \boldsymbol{y}_w, \boldsymbol{y}_l) \in \mathcal{P}_i} \mathcal{L}_{BOPO}(\pi_{\boldsymbol{\theta}}, \boldsymbol{x}, \boldsymbol{y}_w, \boldsymbol{y}_l)$
11:        $\boldsymbol{\theta} \leftarrow \text{Adam}(\boldsymbol{\theta}, \nabla_{\boldsymbol{\theta}} \mathcal{L}(\boldsymbol{\theta}), \eta)$
12:     **end for**
13: **end for**

---

ing factor derived from explicit preferences, which acts as a *natural curriculum*. For different pairs with the same best solution $\boldsymbol{y}_w$ but different suboptimal solutions $\boldsymbol{y}_l$, their preference differences should vary according to explicit preferences. Therefore, it is wise to introduce a scaling factor to adjust the difference in the preference distribution.

By maximizing the log-likelihood of $p_{\boldsymbol{\theta}}(\boldsymbol{y}_w \succ \boldsymbol{y}_l | \boldsymbol{x})$, the model is encouraged to assign higher probabilities to preferred solutions $\boldsymbol{y}_w$ compared with less preferred solutions $\boldsymbol{y}_l$. From Equations (1) to (3), we can derive the BOPO loss function:

$$\mathcal{L}_{BOPO}(\pi_{\boldsymbol{\theta}}, \boldsymbol{x}, \boldsymbol{y}_w, \boldsymbol{y}_l) =$$

$$-\log \sigma \left( \underbrace{\frac{g(\boldsymbol{y}_l)}{g(\boldsymbol{y}_w)}}_{\text{Adaptive Scaling}} \left( \underbrace{\frac{\log \pi_{\theta}(\boldsymbol{y}_w|\boldsymbol{x})}{|\boldsymbol{y}_w|} - \frac{\log \pi_{\theta}(\boldsymbol{y}_l|\boldsymbol{x})}{|\boldsymbol{y}_l|}}_{\text{Average Log-likelihood Difference}} \right) \right). \tag{4}$$

The BOPO training algorithm is presented in Algorithm 1.

**Comparison with Other Losses.** Our BOPO loss differs from existing preference optimization losses in several key aspects. Compared with RLHF (Stiennon et al., 2020), it eliminates the need to train an additional reward model. Compared with DPO (Rafailov et al., 2023), it avoids using a reference model, reducing computational costs. Compared with SimPO (Meng et al., 2024), it incorporates a objective-guided scaling factor without requiring extra hyperparameters, avoiding labor-intensive hyperparameter tuning. Detailed analyses are provided in Appendix C.

**Comparison with RL.** By leveraging preference learning between the best solution and diverse inferior ones, BOPO guides the model toward promising decision trajectories and

discern suboptimal choices, demonstrating its advantage against RL methods. Meanwhile, recognizing that precise rewards (i.e., objective values) in RL are crucial to combinatorial optimization, BOPO introduces an objective-guided scaling factor beyond the standard preference optimization loss function to better distinguish preference differences.

### 4.4. Characteristics of BOPO

In summary, our BOPO has the following characteristics. (1) **Novel training paradigm**: BOPO introduces preference optimization to neural combinatorial optimization as a new training paradigm, featuring two effective problem-awareness components: best-anchored preference pair construction and objective-guided preference optimization loss. (2) **Architecture agnostic**: BOPO is compatible with various models for different problems, achieving high sample efficiency without expensive labels while inheriting the fast inference advantage of neural models. (3) **Outstanding performance**: Compared with existing methods, BOPO surpasses state-of-the-art results on classic COPs, including JSP, TSP, and FJSP.

## 5. Experimental Results

To evaluate the performance of BOPO, we compare it with state-of-the-art NCO methods and strong traditional solvers on typical COP benchmarks with various problem shapes and distributions. Performance evaluation is based on the gap metric $\frac{g(\boldsymbol{y})-g(\boldsymbol{y}^*)}{g(\boldsymbol{y}^*)} \times 100\%$ between the obtained solution $\boldsymbol{y}$ and known optimal solution $\boldsymbol{y}^*$, where a lower gap indicates better performance. The best results are highlighted in **bold**. We also report total solving time for each instance group. Experiments were conducted on a Linux system with an NVIDIA TITAN Xp GPU and an Intel(R) Xeon(R) E5-2680 CPU. Our implementation of BOPO using PyTorch and trained models for each problem are available.[1]

### 5.1. Job-shop Scheduling Problem

**Neural Model.** Each JSP instance is represented as a disjunctive graph, a standard representation for scheduling problems. For details of disjunctive graphs, see Appendix A. We employ a neural model, named MGL, that combines a multi-layer graph attention network (GAT) (Veličković et al., 2018) encoder for computing node embeddings with a long short-term memory (LSTM) based (Hochreiter & Schmidhuber, 1997) context-attention decoder for predicting action probabilities using both embedding and context features. The complete architecture is detailed in Appendix B.

**Training & Test.** For evaluation, we use three standard JSP benchmarks: Taillard's (TA) (Taillard, 1993), Lawrence's

---

[1]https://github.com/L-Z-7/BOPO

(LA) (Lawrance, 1984), and Demirkol's (DMU) (Demirkol et al., 1998). Each benchmark contains 8 different shapes with 10 instances per shape, except LA which has 5 instances per shape. We generate a training dataset of 30000 instances following SLIM (Corsini et al., 2024), consisting of 6 shapes $(n \times m)$ in $\{10 \times 10, 15 \times 10, 15 \times 15, 20 \times 10, 20 \times 15, 20 \times 20\}$ with 5000 instances per shape. During training, we generate additional 100 different instances per shape from the same shape set for validation. We employ the Adam optimizer (Kingma & Ba, 2014) with learning rate $\eta = 0.0002$ and train the neural model for 20 epochs. We set the solution number of hybrid rollout $B = 256$, the number of filtered solutions $K = 16$, batch size of $D = 1$. During inference, we adopt both greedy rollout and sampling rollout with $B'$ solutions.

**Baselines.** We compare BOPO with two categories of approaches: non-constructive methods and constructive methods. (1) Non-constructive methods, which require extensive search time, include both exact solvers and state-of-the-art neural improvement methods. We employ two exact solvers: **Gurobi** and Google **OR Tools**, both with a time limit of 3600 seconds. We also include four RL-based improvement methods: **NLS**$_A$ (Falkner et al., 2022), **L2S** with 500 (L2S500) and 5000 (L2S5k) solutions (Zhang et al., 2024a), and **TGA**500 (Zhang et al., 2024b) with 500 solutions. (2) Constructive methods comprise widely used traditional constructive heuristics and state-of-the-art neural constructive methods, where neural methods adopt both greedy rollout and sampling rollout with $B'$ solutions. For traditional constructive heuristics, we consider three representative traditional Priority Dispatching Rules (PDRs) (Haupt, 1989): **shortest processing time** (SPT), **most operations remaining** (MOR), and **most work remaining** (MWR). The neural constructive baselines include three RL-based methods: **L2D** (Zhang et al., 2020) and **SchN** (Park et al., 2021a), which utilize PPO with different modeling approaches, and **CL** (Iklassov et al., 2023), which incorporates curriculum learning. We also include two state-of-the-art SLL-based baselines, **SLIM** (Corsini et al., 2024) and **SLIM**$_{MGL}$, where SLIM$_{MGL}$ uses our MGL model with SLIM's training algorithm. For a fair comparison, we set its batch size to 16, matching BOPO's setting.

**Results on JSP Benchmarks.** Comparative results on the TA and LA benchmarks are presented in Table 1. Our method achieves the lowest average optimality gap among all constructive methods on all benchmarks, with sampling rollout further enhancing its performance through the exploration of more solutions. Notably, BOPO even surpasses RL-based improvement methods, with the exception of L2S$_{5k}$, where BOPO achieves a comparable gap (7.5% vs. 7.4% on TA) while requiring significantly less computational time (4.8m vs. 4h on TA). Detailed runtime analysis is provided in Appendix D. Compared with SLIM, the current state-

*Table 1.* Average gaps (%) of evaluated methods on JSP benchmarks. "-" indicates unavailable results from the corresponding paper.

| | | Non-constructive | | | | | | Greedy Constructive | | | | | | | | BOPO | Sampling Constructive | | | | | | | |
| | | Exact Solver | | RL-based Improvement | | | | Traditional PDR | | | RL | | | SLL | | | $B'=128$ | | | | | $B'=512$ | | |
| | Shape | Gurobi | OR-Tools | $L2S_{500}$ | $NLS_A$ | $TGA_{500}$ | $L2S_{5k}$ | SPT | MOR | MWR | L2D | SchN | CL | $SLIM_{MGL}$ | SLIM | BOPO | L2D | CL | $SLIM_{MGL}$ | SLIM | BOPO | $SLIM_{MGL}$ | SLIM | BOPO |
|---|---|---|---|---|---|---|---|---|---|---|---|---|---|---|---|---|---|---|---|---|---|---|---|---|
| TA | 15×15 | 0.1 | 0.1 | 9.3 | 7.7 | 8.0 | 6.2 | 25.8 | 20.5 | 19.2 | 26.0 | 15.3 | 14.3 | 13.1 | 13.8 | 13.6 | 17.1 | 9.0 | 8.8 | 7.2 | 7.1 | 7.2 | 6.5 | **6.3** |
| | 20×15 | 3.2 | 0.2 | 11.6 | 12.2 | 9.9 | 8.3 | 32.9 | 23.6 | 23.4 | 30.0 | 19.4 | 16.0 | 16.1 | 15 | 14.3 | 23.7 | 10.6 | 11.0 | 9.3 | 9.0 | 10.4 | 8.8 | **8.3** |
| | 20×20 | 2.9 | 0.7 | 12.4 | 11.5 | 10.0 | 9.0 | 27.8 | 21.7 | 21.8 | 31.6 | 17.2 | 17.3 | 15.3 | 15.2 | 15.1 | 22.6 | 10.9 | 11.1 | 10.0 | 9.8 | 10.0 | **9.0** | 9.1 |
| | 30×15* | 10.7 | 2.1 | 14.7 | 14.1 | 13.3 | 9.0 | 35.1 | 22.7 | 23.7 | 33.0 | 19.1 | 18.5 | 17.7 | 17.1 | 16.6 | 24.4 | 14.0 | 14.0 | 11.0 | 11.0 | 12.2 | 10.6 | **10.3** |
| | 30×20* | 13.2 | 2.8 | 17.5 | 16.4 | 16.4 | 12.6 | 34.4 | 24.9 | 25.2 | 33.6 | 23.7 | 21.5 | 19.3 | 18.5 | 17.1 | 28.4 | 16.1 | 16.3 | 13.4 | 13.3 | 14.9 | 12.7 | **12.2** |
| | 50×15* | 12.2 | 3.0 | 11.0 | 11.0 | 9.6 | 4.6 | 24.1 | 17.3 | 16.8 | 22.4 | 13.9 | 12.2 | 13.4 | 10.1 | 9.8 | 17.1 | 9.3 | 9.2 | 5.5 | 5.8 | 8.2 | **4.9** | 4.9 |
| | 50×20* | 13.6 | 2.8 | 13.0 | 11.2 | 11.9 | 6.5 | 25.6 | 17.7 | 17.9 | 26.5 | 13.5 | 13.2 | 14.0 | 11.6 | 11.8 | 20.4 | 9.9 | 10.6 | 8.4 | 8.0 | 9.8 | 7.6 | **7.4** |
| | 100×20* | 11.0 | 3.9 | 7.9 | 5.9 | 6.4 | 3.0 | 14.4 | 9.2 | 8.3 | 13.6 | 6.7 | 5.9 | 7.4 | 5.8 | 4.9 | 13.3 | 4.0 | 4.8 | 2.3 | 1.8 | 4.4 | 2.1 | **1.4** |
| | Avg | 8.4 | 2.0 | 12.2 | 11.3 | 10.7 | 7.4 | 27.5 | 19.7 | 19.5 | 27.1 | 16.1 | 14.9 | 14.5 | 13.4 | 12.9 | 20.8 | 10.4 | 10.7 | 8.4 | 8.2 | 9.6 | 7.8 | **7.5** |
| LA | 10×5* | 0.0 | 0.0 | 2.1 | - | 2.1 | 1.8 | 14.8 | 16.0 | 16.0 | 14.3 | 12.1 | - | 8.6 | 9.3 | 6.0 | 8.8 | - | 3.7 | 1.9 | 2.7 | 2.5 | **1.1** | 2.1 |
| | 10×10 | 0.0 | 0.0 | 4.4 | - | 1.8 | 0.9 | 15.7 | 18.1 | 12.2 | 23.7 | 11.9 | - | 9.1 | 8.9 | 8.2 | 10.4 | - | 3.5 | 3.1 | 2.3 | 2.4 | 2.5 | **2.1** |
| | 15×5* | 0.0 | 0.0 | 0.0 | - | 0.0 | 0.0 | 14.9 | 3.9 | 5.5 | 7.8 | 2.7 | - | 1.5 | 2.6 | 1.1 | 2.8 | - | 0.0 | 0.0 | 0.0 | 0.0 | 0.0 | 0.0 |
| | 15×10 | 0.0 | 0.0 | 6.4 | - | 3.6 | 3.4 | 28.7 | 23.7 | 17.8 | 27.2 | 14.6 | - | 11.7 | 11.6 | 11.0 | 16.2 | - | 6.3 | 5.2 | 5.8 | 5.6 | 5.0 | **4.9** |
| | 15×15 | 0.0 | 0.0 | 7.3 | - | 5.5 | 5.9 | 24.6 | 18.1 | 18.2 | 27.1 | 16.1 | - | 13.5 | 13.6 | 12.2 | 17.4 | - | 7.1 | 6.8 | 6.5 | 6.7 | 5.6 | **4.9** |
| | 20×5* | 0.0 | 0.0 | 0.0 | - | 0.0 | 0.0 | 13.7 | 3.8 | 5.2 | 6.3 | 3.6 | - | 1.5 | 2.1 | 0.4 | 3.1 | - | 0.5 | 0.0 | 0.0 | 0.0 | 0.0 | 0.0 |
| | 20×10 | 0.0 | 0.0 | 7.0 | - | 5.0 | 2.6 | 33.4 | 20.9 | 17.2 | 24.6 | 15.7 | - | 14.3 | 12.1 | 12.2 | 18.3 | - | 7.9 | 6.9 | 5.9 | 7.1 | 5.6 | **4.6** |
| | 30×10* | 0.0 | 0.0 | 0.2 | - | 0.0 | 0.0 | 13.9 | 6.5 | 8.6 | 8.4 | 3.1 | - | 3.1 | 2 | 2.4 | 6.8 | - | 0.3 | 0.0 | 0.0 | 0.1 | **0.0** | 0.0 |
| | Avg | 0.0 | 0.0 | 3.4 | - | 2.3 | 1.8 | 20.0 | 13.9 | 12.6 | 17.4 | 10.0 | - | 7.9 | 7.8 | 6.7 | 10.6 | - | 3.7 | 3.0 | 2.9 | 3.0 | 2.5 | **2.3** |
| DMU | 20×15 | 5.3 | 1.8 | - | - | - | - | 28.0 | 30.9 | 28.8 | 39.0 | - | - | 17.0 | 18 | 17.5 | 29.3 | 19.4 | 13.7 | 12.0 | 11.2 | 12.7 | 11.3 | **10.4** |
| | 20×20 | 4.7 | 1.9 | - | - | - | - | 31.3 | 27.4 | 27.3 | 37.7 | - | - | 22.6 | 19.4 | 20.3 | 27.1 | 16.0 | 15.3 | 13.5 | 12.7 | 14.1 | 12.3 | **11.8** |
| | 30×15* | 14.2 | 2.5 | - | - | - | - | 31.5 | 37.4 | 32.3 | 42.0 | - | - | 24.1 | 21.8 | 19.1 | 34.0 | 16.5 | 18.4 | 14.4 | 13.9 | 17.5 | 14.0 | **12.9** |
| | 30×20* | 16.7 | 4.4 | - | - | - | - | 34.4 | 34.7 | 31.4 | 39.7 | - | - | 25.6 | 25.7 | 25.6 | 33.6 | 20.2 | 19.0 | 17.1 | 16.5 | 17.8 | 15.8 | **15.5** |
| | 40×15* | 16.3 | 4.1 | - | - | - | - | 24.0 | 36.7 | 27.5 | 35.6 | - | - | 20.1 | 17.5 | 15.9 | 31.5 | 17.6 | 15.8 | 11.7 | 11.4 | 15.3 | 10.9 | **10.9** |
| | 40×20* | 22.5 | 4.6 | - | - | - | - | 37.2 | 37.1 | 32.9 | 39.6 | - | - | 23.5 | 22.2 | 22.3 | 35.8 | 25.6 | 19.8 | 16.0 | 16.7 | 19.0 | **14.8** | 15.9 |
| | 50×15* | 14.9 | 3.8 | - | - | - | - | 24.8 | 35.5 | 28.0 | 36.5 | - | - | 18.2 | 15.7 | 14.5 | 32.7 | 21.7 | 15.6 | 11.2 | 11.2 | 15.3 | 10.6 | **10.4** |
| | 50×20* | 22.5 | 4.8 | - | - | - | - | 30.1 | 37.0 | 30.8 | 39.5 | - | - | 25.8 | 22.4 | 25.2 | 36.1 | 15.2 | 20.8 | 15.8 | 16.5 | 20.0 | **15.0** | 15.5 |
| | Avg | 14.6 | 3.5 | - | - | - | - | 30.2 | 34.6 | 29.9 | 38.7 | - | - | 22.1 | 20.3 | 20.0 | 32.5 | 19.0 | 17.3 | 14.0 | 13.8 | 16.5 | 13.1 | **12.9** |

of-the-art SLL-based constructive method, BOPO, which employs the efficient MGL model, achieves both reduced parameter count and computational overhead (detailed in Appendix B). More significantly, when evaluated against SLIM_MGL using the identical model, the performance disparity increases markedly across all scenarios, underscoring the fundamental advantage of the proposed training paradigm over the SLL counterpart. Furthermore, BOPO exhibits superior generalization performance on out-of-distribution problem shapes (marked by *).

## 5.2. Traveling Salesman Problem

**Neural Model.** BOPO adopts the same model as the typical POMO (Kwon et al., 2020), comprising an encoder with 6 Transformer layers and a decoder with a multi-head attention layer. To further demonstrate the universality of BOPO, it also employs the state-of-the-art INViT (Fang et al., 2024) model.

**Training & Test.** Following the NCO literature, we evaluate the proposed method on randomly generated instances with $n = 20/50/100$ (denoted as TSP20/50/100), using models trained on corresponding problem shapes. Training instances are generated randomly from an uniform distribution. Additionally, we assess BOPO's generalization capability on the out-of-distribution TSPLIB benchmark (Reinelt, 1991), and randomly generated instances with four different distributions, including uniform, cluster, explosion, and implosion distributions. We adopt the same hyperparameter configuration as POMO and INViT, respectively. During training, the number of filtered solutions is set to $K = 8$. For TSP20 and TSP50, we set the hybrid rollout solution number $B = 128$ and batch size $D = 64$. For TSP100, we set $B = 256, K = 16$ and $D = 48$ due to the memory limit. Specifically, we adopt BOPO to train INViT-2V, but due to memory constraints, we set $D = 1, B = 64$ (half of batch size in INViT), with $K = 8$ and double epochs. For inference, we adopt greedy rollout with multiple start nodes and ×8 instance augmentation (denoted as aug.), consistent with POMO.

**Baselines.** We assess BOPO against both traditional methods and neural constructive methods. The traditional methods include exact solver Concorde and Gurobi, and LKH3, a powerful problem-specific heuristic. The neural constructive methods include POMO (Kwon et al., 2020), a widely-adopted RL-based backbone for advanced methods; DABL (Yao et al., 2024), a state-of-the-art SL-based method with data augmentation for routing problems; INViT (Fang et al., 2024), a state-of-the-art RL-based method; and SLIM (Corsini et al., 2024), a state-of-the-art SLL-based method applied to the POMO model.

**Results on TSP.** Results based on POMO are presented in Table 2 and Table 3. As shown in Table 2, BOPO outperforms other neural baselines (except DABL on TSP50) while delivering competitive solutions against traditional solvers, despite the latter requiring substantially more computational time. Compared with DABL which requires expensive labeled optimal solutions for SL, BOPO achieves

*Table 2.* Results on 1000 uniformly generated TSP instances.

| Method | TSP20 Obj.↓ | Gap↓ | Time↓ | TSP50 Obj.↓ | Gap↓ | Time↓ | TSP100 Obj.↓ | Gap↓ | Time↓ |
|---|---|---|---|---|---|---|---|---|---|
| Concorde | 3.83 | 0.00 | 5m | 5.69 | 0.00 | 13m | 7.75 | 0.00 | 1h |
| Gurobi | 3.83 | 0.00 | 7s | 5.69 | 0.00 | 2m | 7.75 | 0.00 | 17m |
| LKH3 | 3.83 | 0.00 | 42s | 5.69 | 0.00 | 6m | 7.75 | 0.00 | 25m |
| POMO | 3.83 | 0.04 | 3.3s | 5.70 | 0.21 | 6.4s | 7.80 | 0.46 | 11.4s |
| DABL | 3.83 | 0.01 | 3.3s | 5.69 | 0.04 | 6.4s | 7.77 | 0.29 | 11.4s |
| SLIM | 3.85 | 0.22 | 3.3s | 5.78 | 1.51 | 6.4s | 8.18 | 5.51 | 11.4s |
| BOPO | 3.83 | 0.02 | 3.3s | 5.70 | 0.14 | 6.4s | 7.78 | 0.37 | 11.4s |
| POMO (aug.) | 3.83 | 0.00 | 3.6s | 5.69 | 0.03 | 6.6s | 7.77 | 0.14 | 18.1s |
| DABL (aug.) | **3.83** | **0.00** | 3.6s | **5.69** | **0.00** | 6.6s | 7.75 | 0.05 | 18.1s |
| SLIM (aug.) | 3.84 | 0.01 | 3.6s | 5.70 | 0.15 | 6.6s | 7.84 | 1.17 | 18.1s |
| BOPO (aug.) | **3.83** | **0.00** | 3.6s | 5.69 | 0.01 | 6.6s | **7.75** | **0.04** | 18.1s |

*Table 3.* Generalization on TSPLIB with various problem shapes.

| Method | $n < 100$ (6 instances) Obj.↓ | Gap↓ | Time↓ | $100 \le n < 200$ (21 instances) Obj.↓ | Gap↓ | Time↓ | $200 \le n < 500$ (16 instances) Obj.↓ | Gap↓ | Time↓ | $500 \le n < 1k$ (6 instances) Obj.↓ | Gap↓ | Time↓ |
|---|---|---|---|---|---|---|---|---|---|---|---|---|
| POMO (aug.) | 6.26 | 2.36 | 0.15s | 6.75 | 3.08 | 0.27s | 10.63 | 14.81 | 0.95s | 16.22 | 30.14 | 4.6s |
| SLIM (aug.) | 6.19 | 1.36 | 0.15s | 6.88 | 5.24 | 0.27s | 10.82 | 16.99 | 0.95s | 19.40 | 55.57 | 4.6s |
| BOPO (aug.) | **6.19** | **1.26** | 0.15s | **6.72** | **2.55** | 0.27s | **10.21** | **10.41** | 0.95s | **15.29** | **22.44** | 4.6s |

comparable performance, demonstrating its superiority. Notably, our BOPO achieves superior performance over POMO on TSP100, even with fewer training epochs (700 vs. 2000). Generalization results on TSPLIB presented in Table 3 demonstrate BOPO's significant advantages over other neural baselines when generalizing to out-of-distribution instances. Moreover, results based on INViT's generalization to other distributions are shown in Figure 2, showing consistent superiority when implemented on different models.

## 5.3. Flexible Job-shop Scheduling Problem

**Neural Model.** For an FJSP instance represented as a disjunctive graph, we adopt the MGL model (see Appendix B) to compute action probabilities, which is similar to the model for JSP.

**Training & Test.** For evaluation, we use FJSP instances from the LA benchmark (Lawrance, 1984). The benchmark includes *e-data*, *r-data*, and *v-data*, where each operation can be allocated to 1-2 machines, 1-3 machines, and 1-$m$ machines, respectively. Following Song et al. (2023), we generate 25000 FJSP instances with 5 different shapes: $\{10 \times 5 \times 5, 10 \times 10 \times 10, 15 \times 10 \times 10, 20 \times 5 \times 5, 20 \times 10 \times 7\}$. For each shape, 2500 instances belong to *r-data* and 2500 to *v-data*. We set $B = 128$, $K = 16$, and $D = 1$ during training, and employ both greedy rollout and sampling rollout with $B'$ solutions during inference.

**Baselines.** For FJSP, we compare our proposal with both representative traditional PDRs and state-of-the-art RL-based constructive methods: DNN (Yuan et al., 2024) using the actor-critic framework, HG (Song et al., 2023) utilizing heterogeneous graphs for instance representation, and RS

*Table 4.* Average gaps (%) on FJSP benchmarks.

| Benchmarks | Greedy Constructive Traditional PDR SPT | MOR | MWR | RL DNN | HG | RS | BOPO | Sampling Constructive $B'$=100 HG | RS | BOPO | $B'$=256 BOPO | $B'$=512 BOPO |
|---|---|---|---|---|---|---|---|---|---|---|---|---|
| LA(e-data) | 26.1 | 17.7 | 20.5 | 15.5 | 15.5 | 13.2 | 14.5 | 8.2 | 6.9 | 6.1 | 5.4 | **5.0** |
| LA(r-data) | 28.7 | 14.4 | 17.8 | 12.1 | 11.2 | 9.6 | 8.4 | 5.8 | 4.7 | 4.0 | 3.6 | **3.4** |
| LA(v-data) | 17.8 | 6.0 | 6.6 | 5.4 | 4.3 | 3.8 | 1.8 | 1.4 | 0.8 | 0.6 | 0.5 | **0.4** |

*Table 5.* Average gaps (%) of various preference pair construction methods on the DMU benchmark.

| Shape | w/o Hybrid Rollout Sampling Rollout | w/o Uniform Filtering Random | Top-$K$ | Bottom-$K$ | w/o Best-anchored Pairing Full Permutation Pairing | BOPO |
|---|---|---|---|---|---|---|
| 20x15 | 11.6 | 12.0 | 12.8 | 12.5 | 11.9 | **10.9** |
| 20x20 | 13.0 | 12.9 | 14.5 | **12.5** | 13.3 | **12.5** |
| 30x15 | 15.2 | 14.0 | 15.9 | 14.6 | 15.5 | **13.3** |
| 30x20 | 16.7 | 16.4 | 18.5 | 16.7 | 16.6 | 15.8 |
| 40x15 | 12.1 | 11.7 | 13.2 | 11.8 | **11.2** | 11.2 |
| 40x20 | 17.5 | **16.1** | 18.7 | 16.4 | 16.5 | **16.1** |
| 50x15 | 12.1 | 12.1 | 12.8 | 11.4 | 11.2 | **10.9** |
| 50x20 | 17.1 | **16.3** | 18.8 | 16.6 | 16.4 | **16.3** |
| Avg | 14.4 | 14.0 | 15.6 | 14.1 | 14.1 | **13.3** |

(Ho et al., 2024) employing residual scheduling to remove finished operations.

**Results on FJSP Benchmarks.** As shown in Table 4, with greedy rollout, BOPO significantly outperforms most baselines, only marginally falling behind RS on LA e-data. When all methods use sampling rollout with 100 solutions, the proposed method even achieves the best performance across all cases. Notably, increasing the number of sampled solutions consistently reduces the optimality gap.

## 5.4. Ablation Study

**Higher Sample Efficiency of the BOPO Training Paradigm.** We compare BOPO with two representative training paradigms: RL and SLL. For JSP, we compare BOPO with RL-based PPO and SLL-based SLIM$_{MGL}$, all using the MGL model and identical training settings. For TSP, we compare BOPO with RL-based POMO and SLL-based SLIM, all using the POMO model and identical training settings. The training curves in Figure 3 demonstrate our proposal's higher sample efficiency, achieving lower optimality gaps than both baselines with the same number of training instances, with the advantage being more pronounced when training data is limited.

**Effectiveness of the Best-anchored Preference Pair Construction Method.** To validate our three-step preference pair construction method, we replace each step with a simpler alternative. We substitute sampling rollout for hybrid rollout, replace uniform filtering with random, top-$K$, or bottom-$K$ filtering, and use full permutation pairing instead of best-anchored pairing while maintaining the same total number of pairs. As shown in Table 5, BOPO's performance deteriorates without (w/o) any of these components,

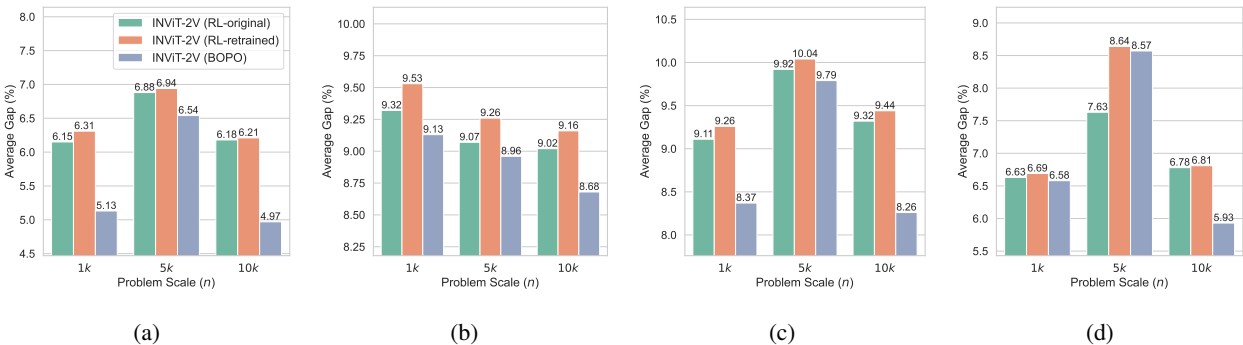

*Figure 2.* Performance on TSP with different problem scales and distributions: (a) uniform, (b) cluster, (c) explosion, and (d) implosion. RL-original denotes the results reported in the INViT paper; RL-retrained denotes the results retrained with the same batch size as BOPO.

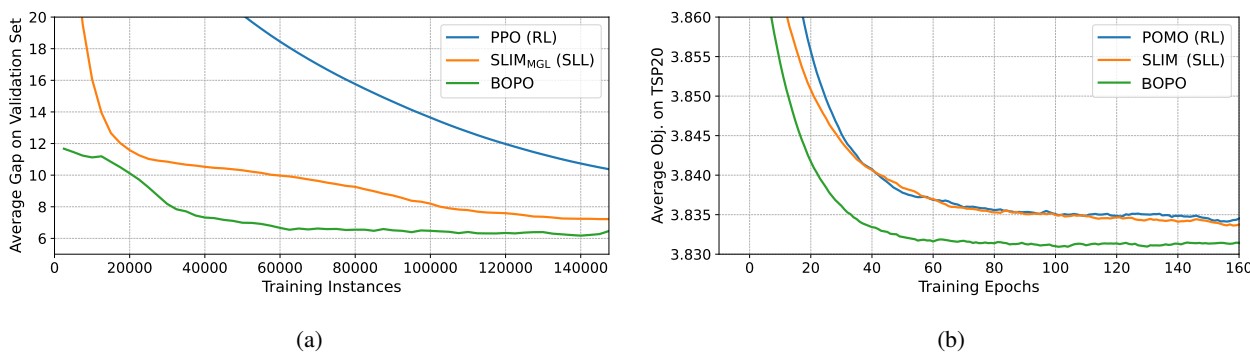

*Figure 3.* Training curves of different training paradigms for: (a) JSP and (b) TSP.

demonstrating the necessity of our design. Notably, top-$K$ filtering leads to significantly worse performance, highlighting the importance of filtering worse solutions to create sufficient preference differences in pair construction. Additional analyses of hybrid rollout effectiveness are provided in Appendix E.

**Superiority of the Objective-guided Preference Optimization Loss.** To evaluate our proposed loss with a preference-based scaling factor, we compare it with other preference optimization losses: the classic DPO loss ($\mathcal{L}_{DPO}$) (Rafailov et al., 2023), the popular SimPO loss ($\mathcal{L}_{SimPO}$) (Meng et al., 2024), and a variant of BOPO loss without the scaling factor ($\mathcal{L}_{BOPO-}$). Complete loss formulations are provided in Appendix C. For DPO, which requires a reference model to prevent excessive policy deviation, we use the old model from 10 episodes prior, similar to PPO, with the common hyperparameter setting $\beta = 0.1$. For SimPO, we follow their standard parameters with $\beta = 2$ and $\gamma = 1$. As shown in Table 6, BOPO achieves the best performance across all benchmarks for both $B' = 128$ and $B' = 512$, with particularly significant improvements on the DMU benchmark, demonstrating the effectiveness of our loss design and its preference-based scaling factor. It is worth noting that SimPO performs poorly on DMU, where

*Table 6.* Average gaps (%) of various loss functions for JSP.

| Benchmark | $\mathcal{L}_{SimPO}$ | $\mathcal{L}_{DPO}$ | $\mathcal{L}_{BOPO-}$ | $\mathcal{L}_{BOPO}$ | $\mathcal{L}_{SimPO}$ | $\mathcal{L}_{DPO}$ | $\mathcal{L}_{BOPO-}$ | $\mathcal{L}_{BOPO}$ |
|---|---|---|---|---|---|---|---|---|
| | | $B' = 128$ | | | | $B' = 512$ | | |
| TA | 8.5 | 8.7 | 8.5 | 8.2 | 7.7 | 7.8 | 7.6 | **7.5** |
| LA | 2.9 | 2.9 | 2.9 | 2.9 | 2.4 | 2.5 | 2.4 | **2.3** |
| DMU | 15.2 | 14.5 | 14.1 | 13.8 | 14.1 | 13.6 | 13.2 | **12.9** |

the test distribution diverges from training. This hints at potential overfitting caused by its target margin term $\gamma$.

### 5.5. Hyperparameter Study

BOPO has two crucial hyperparameters: the solution number of hybrid rollouts $B$ and the number of filtered solutions $K$. We analyze their individual effects by varying $B \in \{32, 64, 128, 256, 512\}$ and $K \in \{4, 8, 16, 32\}$. Additionally, we explore their interaction by maintaining a fixed ratio $B/K = 16$ while scaling both parameters $K \times B \in \{4 \times 64, 8 \times 128, 16 \times 256, 32 \times 512\}$.

**Effect of the Solution Number of Hybrid Rollouts.** As shown in Figure 4a, increasing the number of sampled solutions during training, i.e., larger $B$, improves solution quality by enhancing the probability of collecting higher-quality

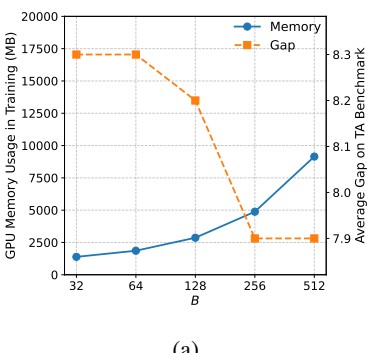 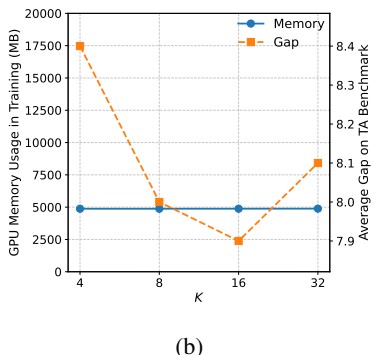 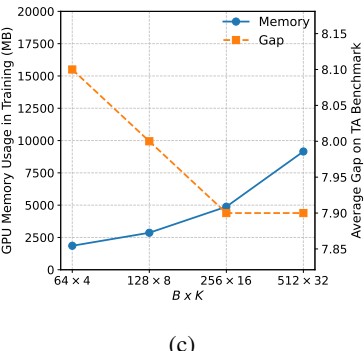

(a)        (b)        (c)

*Figure 4.* GPU memory usage in training and average gap (%) on TA benchmark for: (a) varying $B$, (b) varying $K$, and (c) varying $B \times K$.

solutions. However, GPU memory consumption grows with $B$ due to parallel computation, raising costs. While the performance improves significantly up to $B = 256$, further increases yield diminishing returns despite rising computational costs, making $B = 256$ a reasonable choice. We also evaluate our method on TSP20/50 with $B = 20/50$, the performance remains comparable to $B = 128$ (detailed in Appendix F). This suggests that for small-scale problems, the optimal $B$ is low since the model can efficiently sample high-quality solutions.

**Effect of the Number of Filtered Solutions.** The number of filtered solutions $K$ determines the number of preference pairs, making it a critical parameter. As shown in Figure 4b, while increasing $K$ generates more preference pairs, it also increases the similarity among solutions, as they are more likely to come from the same local region. Our experiments show that $K = 16$ achieves the best performance, with either larger or smaller values leading to performance degradation. This suggests that a moderate $K$ value balances the trade-off between sufficient training data and solution diversity.

**Effect of Interaction Between Number of Rollouts and Filtered Solutions.** $B$ and $K$ are interdependent parameters, as $B$ affects the uniform filtering step size $\lfloor B/K \rfloor$, which influences the similarity between solutions. As shown in Figure 4c, we maintain a fixed step size of 16 while scaling $B$ and $K$ proportionally. The performance improves until $B \times K$ reaches $256 \times 16$, after which larger values yield minimal gains despite increased memory costs. This further validates our choice of $B = 256$ and $K = 16$ as recommended parameters.

## 6. Conclusion

In this work, we present BOPO, a preference optimization-based training paradigm for neural combinatorial optimization. By introducing best-anchored preference pair construction and a novel objective-guided pairwise loss function for COPs, our proposal achieves higher sample efficiency

than mainstream RL and recent SLL paradigms. Extensive experiments on JSP, TSP, and FJSP demonstrate BOPO's superior performance over state-of-the-art neural constructive methods, while requiring significantly less time to deliver solutions competitive with traditional problem-specific iterative heuristics. The proposed method requires neither expensive labels nor specialized design of the Markov Decision Process, making it easy to use in practice. More importantly, it establishes a general training paradigm that can be readily applied to various neural models for solving different COPs.

Although BOPO demonstrates superior sample efficiency compared with RL and SLL, one limitation is that it still requires a relatively large number of rollout solutions, similar to SLIM. This incurs moderate costs in collecting high-quality solutions for effective model learning. In future work, we will explore efficient ways to obtain high-quality solutions that facilitate model training. One promising direction is to leverage problem invariance and solution symmetry to efficiently generate diverse high-quality training data. Another direction is to efficiently enhance solution quality by incorporating problem-specific heuristics during training, providing better learning signals for the model.

## Acknowledgements

This work is supported by the National Natural Science Foundation of China (62472461) and the Guangdong Basic and Applied Basic Research Foundation (2024A1515010871, 2025A1515010129).

## Impact Statement

This paper introduces research aimed at advancing machine learning, poised to facilitate industries and enhance decision-making. However, its adoption must be accompanied by careful consideration of ethical, societal, and environmental implications to ensure responsible and equitable use.

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

*Table 7.* The state features $\boldsymbol{s}_j \in \mathbb{R}^{15}$ that describes the information of an operation $O_j$ for JSP.

| ID | Description |
|---|---|
| 1 | The processing time $t_j$ of the operation. |
| 2 | The completion of job $J_i$ up to $O_j$: $\sum_{j'=l_{i1}}^{j} t_{j'} / \sum_{j' \in \mathcal{O}_i} t_{j'}$. |
| 3 | The completion of job $J_i$ after $O_j$: $\sum_{j'=j+1}^{l_{im}} t_{j'} / \sum_{j' \in \mathcal{O}_i} t_{j'}$. |
| 4-6 | The $1^{st}$, $2^{nd}$, and $3^{rd}$ quartile among processing times of operations on job $J_i$. |
| 7-9 | The $1^{st}$, $2^{nd}$, and $3^{rd}$ quartile among processing times of operations on machine $M_j$. |
| 10-12 | The difference between $t_j$ and feature 4 6. |
| 13-15 | The difference between $t_j$ and feature 7 9. |

*Table 8.* The context features $\boldsymbol{c}_i \in \mathbb{R}^{11}$ that describes the status of a job $J_i$ within a partial solution $\boldsymbol{y}_{<t}$ at step $t$ for JSP.

| ID | Description |
|---|---|
| 1 | $C(J_i)$ minus the completion time of machine $M_{(i,t)}$ |
| 2 | $C(J_i)$ divided by the makespan of partial solution $C(\boldsymbol{y}_{<t})$. |
| 3 | $C(J_i)$ minus the average completion time of all jobs. |
| 4-6 | The difference between $C(J_i)$ and the $1^{st}$, $2^{nd}$, and $3^{rd}$ quartile computed among the completion time of all jobs. |
| 7 | The completion time of machine $M_{(i,t)}$ divided by the makespan of the partial solution $C(\boldsymbol{y}_{<t})$. |
| 8 | The completion time of machine $M_{(i,t)}$ minus the average completion of all machines. |
| 9-11 | The difference between the completion of $M_{(i,t)}$ and the $1^{st}$, $2^{nd}$, and $3^{rd}$ quartile computed among the completion time of all machines. |

# A. Formalization of Problems

## A.1. Job-shop Scheduling Problems

JSP consists of a set of jobs $\mathcal{J} = \{J_1, \cdots, J_n\}$, a set of operations $\mathcal{O} = \{O_1, \cdots, O_l\}$, and a set of machines $\mathcal{M} = \{M_1, \cdots, M_m\}$. Each job $J_i \in \mathcal{J}$ is composed of a sequence of $m$ operations $(O_{l_{i1}}, \cdots, O_{l_{im}})$, where $l_{ij} \in \{1, \cdots, l\}$. It must be completed sequentially in a strict order. Each operation $O_j \in \mathcal{O}$ must be performed on a specific machine $M_j \in \mathcal{M}$ continuously for $t_j$ seconds, and each machine can only process one operation at a time. For convenience, we define assignable operations at step $t$ as $\mathcal{O}_t$, the pending operation of job $J_i$ at step $t$ as $O_{(i,t)}$, and the corresponding machine of $O_{(i,t)}$ as $M_{(i,t)}$. Once a scheduling plan is determined, the completion time $C(O_j)$ of each operation $O_j$ is decided accordingly, resulting in the maximum completion time $C(\boldsymbol{y}) = \max_{i \in \{1,\ldots,n\}} C(J_i)$ of all jobs (i.e., makespan), where $C(J_i) = \max_{j \in \{l_{i1}, \ldots, l_{im}\}} C(O_j)$ represents the completion time of job $J_i$. Makespan is typically the objective to be minimized in JSP.

**Disjunctive Graph**. JSP can be represented using a disjunctive graph $G = (V, A, E)$. In this graph, the node set $V = \{O_j \mid O_j \in \mathcal{O}\}$ represents operations, the directed edge set $A = \{O_{l_{ij}} \to O_{l_{i(j+1)}} \mid O_{l_{ij}}, O_{l_{i(j+1)}} \in \mathcal{O}\}$ indicates precedence constraints between successive operations ($O_{l_{ij}} \to O_{l_{i(j+1)}}$), and the disjunctive (undirected) edge set $E$ connects operations performed on the same machine. A feasible solution is obtained by assigning directions to the undirected edges in $E = \{O_j \leftrightarrow O_{j'} \mid O_j, O_{j'} \in \mathcal{O}, M_j = M_{j'}\}$, resulting in a directed acyclic graph.

**Features of JSP.** Following previous works (Zhang et al., 2020; Corsini et al., 2024), we define two types of features. The first type consists of static state features $\boldsymbol{s}_j$ assigned to each operation $O_j$ in the node set $V$, while the second type comprises contextual features $\boldsymbol{c}_i$ based on job $J_i$ information at step $t$. The state features $\boldsymbol{s}_j$ are fed into the encoder to compute embeddings for each node (operation), while the contextual features $\boldsymbol{c}_i$ are combined with these embeddings to jointly contribute to the prediction. Table 7 details the state features $\boldsymbol{s}_j \in \mathbb{R}^{15}$, and Table 8 details the contextual features $\boldsymbol{c}_i \in \mathbb{R}^{11}$.

## A.2. Traveling Salesman Problem

Two-dimensional Euclidean TSP, which is discussed in this paper, involves $n$ nodes, where each node $i \in \{1, \ldots, n\}$ is represented by a two-dimensional coordinate, forming a fully connected graph. The distance $C(i, j)$ between nodes $i$ and $j$

*Table 9.* The state features $\boldsymbol{s}_{jk} \in \mathbb{R}^{15}$ that describes the information of an operation-machine node $(O_j, M_k)$ for FJSP.

| ID | Description |
|---|---|
| 1 | The processing time $t_{jk}$ of the node $(O_j, M_k)$. |
| 2 | The average completion of job $J_i$ up to $O_j$: $\sum_{j'=l_{i1}}^{j} t_{j'} / \sum_{j' \in \mathcal{O}_i} t_{j'}$. |
| 3 | The average completion of job $J_i$ after $O_j$: $\sum_{j'=j+1}^{l_{im}} t_{j'} / \sum_{j' \in \mathcal{O}_i} t_{j'}$. |
| 4-6 | The $1^{st}$, $2^{nd}$, and $3^{rd}$ quartile among processing times of operations on job $J_i$. |
| 7-9 | The $1^{st}$, $2^{nd}$, and $3^{rd}$ quartile among processing times of operations on machine $M_k$. |
| 10-12 | The difference between $t_j$ and feature 4-6. |
| 13-15 | The difference between $t_j$ and feature 7-9. |

is calculated by their coordinates. The objective of TSP is to find the shortest Hamiltonian cycle $g(\boldsymbol{y}) = \sum_{j=1}^{n} C(y_j, y_{j+1})$, where $\boldsymbol{y} = (y_1, \ldots, y_n, y_1), y_j \in \{1, \ldots, n\}$ visits each node exactly once and returns to the starting node.

**Features of TSP.** Similar to Kool et al. (2019), the input of TSP consists of $n$ nodes with 2-dimensional features. At each decoding step, the context embedding $\boldsymbol{h}_c$ is defined as the concatenation of embeddings from the first and last visited nodes. The coordinates of each instance are sampled from a uniform distribution on $[0, 1]^2$.

### A.3. Flexible Job-shop Scheduling Problem

FJSP, a variant of JSP, better reflects real-world scenarios. Unlike JSP, FJSP allows each operation $O_j \in \mathcal{O}$ to choose from multiple candidate machines $\mathcal{M}_j \subseteq \mathcal{M}$, rather than being restricted to a specific one. We redefine the processing time of $O_j$ in machine $M_k \in \mathcal{M}_j$ as $t_{jk}$, and denote the $t_j = \frac{1}{|\mathcal{M}_j|} \sum_{k=1}^{|\mathcal{M}_j|} t_{jk}$ as the average processing time of operation $O_j$. This flexibility significantly increases decision-making complexity, leading to a denser disjunctive graph.

**Disjunctive Graph for FJSP**. To simplify the disjunctive graph for FJSP, we introduce operation-machine nodes, where an operation is decomposed into multiple operation-machine nodes. Each operation-machine node is similar to a node in JSP and can be treated as an action. Specifically, the disjunctive graph is formalized as $G = (V, A, E, U)$, where $V = \{(O_j, M_k) \mid O_j \in \mathcal{O}, M_k \in \mathcal{M}_j\}$ represents the set of all operation-machine pairs, $A = \{(O_{l_{ij}}, M_k) \rightarrow (O_{l_{i(j+1)}}, M_{k'}) \mid O_{l_{ij}}, O_{l_{i(j+1)}} \in \mathcal{O}; M_k \in \mathcal{M}_{l_{ij}}; M_{k'} \in \mathcal{M}_{l_{i(j+1)}}\}$ denotes the directed edge set, $E = \{(O_j, M_k) \leftrightarrow (O_{j'}, M_k) \mid O_j, O_{j'} \in \mathcal{O}; M_k \in \mathcal{M}_j \cap \mathcal{M}_{j'}\}$ denotes the disjunctive (undirected) edge set, and $U = \{(O_j, M_k) \leftrightarrow (O_j, M_{k'}) \mid O_j \in \mathcal{O}, ; M_k, M_{k'} \in \mathcal{M}_j\}$ connects all operation-machine nodes belonging to the same operation.

**Features of FJSP.** Similar to JSP, we define two types of features for FJSP. To accommodate operation-machine nodes used in FJSP, we split the context features into job context features $\boldsymbol{c}_i^J$ and machine context features $\boldsymbol{c}_k^M$, where $\boldsymbol{c}_i^J$ represents the context features of job $J_i$, and $\boldsymbol{c}_k^M$ represents the context features of machine $M_k$. Similarly, the static state features $\boldsymbol{s}_j \rightarrow \boldsymbol{s}_{jk}$ are modified from describing operation $O_j$ to describing the operation-machine nodes $(O_j, M_k)$. Table 9 details the state features $\boldsymbol{s}_{jk} \in \mathbb{R}^{15}$, and Table 10 details the contextual features $\boldsymbol{c}_i^J, \boldsymbol{c}_k^M \in \mathbb{R}^5$.

## B. Neural Model for Scheduling Problems

For scheduling problems, we design an efficient neural model named MGL, which combines a multi-layer graph attention network (GAT) (Veličković et al., 2018) encoder with a long short-term memory (LSTM) based (Hochreiter & Schmidhuber, 1997) context attention decoder.

### B.1. Neural Model for JSP

**Encoder.** Since the disjunctive graph contains two types of edges: directed edges related to jobs, and disjunctive edges (undirected) related to machines, we treat it as a two-layer graph to better distinguish between these two edge types, i.e., $G_{job} = (V, A), G_{mac} = (V, E)$, and $G = G_{job} \cup G_{mac}$. One layer contains only directed edges, while the other contains only disjunctive edges. To process this structure, we introduce a multi-layer GAT as the encoder, where each layer can be

*Table 10.* The job context features $\boldsymbol{c}_i^J \in \mathbb{R}^5$ and the machine context features $\boldsymbol{c}_k^M \in \mathbb{R}^5$ that describe the status of a job $J_i$ and a machine $M_k$ within a partial solution $\boldsymbol{y}_{<t}$ at step $t$ for JSP.

| Job ID | Description |
|---|---|
| 1 | $C(J_i)$ divided by the makespan of partial solution $C(\boldsymbol{y}_{<t})$. |
| 2 | $C(J_i)$ minus the average completion time of all jobs. |
| 3-5 | The difference between $C(J_i)$ and the $1^{st}$, $2^{nd}$, and $3^{rd}$ quartile computed among the completion time of all jobs. |

| Machine ID | Description |
|---|---|
| 1 | The completion time of machine $M_k$ divided by the makespan of the partial solution $C(\boldsymbol{y}_{<t})$. |
| 2 | The completion time of machine $M_k$ minus the average completion of all machines. |
| 3-5 | The difference between the completion of $M_k$ and the $1^{st}$, $2^{nd}$, and $3^{rd}$ quartile computed among the completion time of all machines. |

considered a standard 2-head GAT. The computation for an $N$-layer $\text{GAT}^N$ is as follows:

$$\text{GAT}^N(\boldsymbol{x}, G_1, \ldots, G_N) = [\sigma(\text{GAT}_1(\boldsymbol{x}, G_1))||\cdots||\sigma(\text{GAT}_N(\boldsymbol{x}, G_N))].$$

In our encoder, we stack two 2-layer GAT as follows to embed $\boldsymbol{e}_j$ of operation $O_j$:

$$\boldsymbol{e}_j = [\boldsymbol{s}_j||\sigma(\text{GAT}^2_{second}([\boldsymbol{s}_j||\sigma(\text{GAT}^2_{first}(\boldsymbol{s}_j, G_{job}, G_{mac}))], G_{job}, G_{mac}))].$$

**Decoder.** At step $t$, we use the embedding $\boldsymbol{e}_{y_{t-1}}$ of the operation selected at step $t-1$ as input to the LSTM, which computes the query $\boldsymbol{q}_t$ as follows:

$$\boldsymbol{q}_t = \text{LN}(\text{LSTM}(\sigma(\boldsymbol{e}_{y_{t-1}} \cdot W_1))) \cdot W_2.$$

For each operation $O_j$, we concatenate context feature $\boldsymbol{c}_i$ of its job $J_i$ with the embedding $\boldsymbol{e}_j$ to obtain the key $\boldsymbol{k}_{t,j}$ as follows:

$$\boldsymbol{k}_{t,j} = [\sigma(\boldsymbol{c}_i \cdot W_3)||\boldsymbol{e}_j] \cdot W_4.$$

Finally, the query $\boldsymbol{q}_t$ attends to the keys $\boldsymbol{k}_{t,j}$ of assignable operations, computing the attention to generate the policy distribution for action selection:

$$\pi(O_j|t) = \frac{\exp(\boldsymbol{q}_t \cdot \boldsymbol{k}_{t,j}^\top)}{\sum_{j' \in \mathcal{O}_t} \exp(\boldsymbol{q}_t \cdot \boldsymbol{k}_{t,j'}^\top)}.$$

### B.2. Neural Model for FJSP

**Encoder for FJSP.** We also employ the MGL model for FJSP. With the introduction of operation-machine nodes and the operation-related edge set $U$, we redefine $G_{opr} = (V, U)$ and $G = G_{job} \cup G_{opr} \cup G_{mac}$. The following describes the embedding $\boldsymbol{e}_{jk}$ of a 3-layer GAT to node $(O_j, M_k)$:

$$\boldsymbol{e}_{jk} = [\boldsymbol{s}_{jk}||\sigma(\text{GAT}^3_{second}([\boldsymbol{s}_{jk}||\sigma(\text{GAT}^3_{first}(\boldsymbol{s}_{jk}, G_{job}, G_{mac}, G_{opr}))], G_{job}, G_{mac}, G_{opr}))].$$

**Decoder for FJSP.** For operation-machine node $(O_j, M_k)$ of job $J_i$, the key $\boldsymbol{k}_{t,j,k}$ is modified as:

$$\boldsymbol{k}_{t,j,k} = [\sigma(\boldsymbol{c}_i^J \cdot W_3)||\sigma(\boldsymbol{c}_k^M \cdot W_4)||\boldsymbol{e}_{jk}] \cdot W_5.$$

Due to the fact that query $\boldsymbol{q}_t$ is independent of operation-machine nodes, the policy distribution for nodes $(O_j, M_k)$ is:

$$\pi((O_j, M_k)|t) = \frac{\exp(\boldsymbol{q}_t \cdot \boldsymbol{k}_{t,j,k}^\top)}{\sum_{O_{j'} \in \mathcal{O}_t, M_{k'} \in \mathcal{M}_{j'}} \exp(\boldsymbol{q}_t \cdot \boldsymbol{k}_{t,j',k'}^\top)}.$$

*Table 11.* MGL's efficiency compared with SLIM's model.

| Model | Parameters | Memory Usage (MB) During Training | Training Time (ms) per Instance | Inference Time (ms) per Instance |
|-------|-----------|------------------------------------|----------------------------------|-----------------------------------|
| MGL | 351.23K | 262.19 | 1395.99 | 800.04 |
| GAT-MHA | 376.96K (+25.73K) | 7110.69 (27x) | 1785.7 (+27.9%) | 851.81 (+6.4%) |

### B.3. Comparison with Neural Model Used in SLIM

Our MGL is compared with the neural model used in SLIM (Corsini et al., 2024), denoted as GAT-MHA. The main distinction between these models lies in their decoders, significantly impacting memory consumption and computational efficiency, as analyzed in Table 11. All data are collected from a $15 \times 15$ instance, with the solution size $B$ set to 256.

The multi-head attention (MHA) module in GAT-MHA introduces additional weight matrices that contribute to its higher parameter count. In contrast, LSTM in MGL achieves efficiency through recurrent weight sharing. GAT-MHA requires 32.8% more memory during forward propagation than MGL. Moreover, GAT-MHA's backward pass memory usage is **27×** **higher** than MGL's, primarily due to MHA's gradient computation needs: *Intermediate Activation Storage* and *Gradient Scaling with Heads*. For the former, MHA must retain attention score matrices and head-specific outputs during forward pass for gradient computation, while LSTM's recurrent nature minimizes intermediate storage. For the latter, MHA's memory overhead scales linearly with the number of attention heads, as gradients for each head's parameters are stored separately.

## C. Comparison with Other Loss Functions

### C.1. Formulations of Loss Functions

In SLIM (Corsini et al., 2024), the locally optimal solution $y_o$ is treated as a pseudo-label, and the model is trained using cross-entropy loss. The loss function of SLIM can be expressed as::

$$\mathcal{L}_{SLIM}(\pi_{\boldsymbol{\theta}}|\boldsymbol{x}, \boldsymbol{y}_o) = -\frac{1}{|\boldsymbol{y}_o|} \log \pi_{\boldsymbol{\theta}}(\boldsymbol{y}_o|\boldsymbol{x}). \tag{5}$$

This formulation effectively maximizes the average log-likelihood of the locally optimal solution. In contrast to SLIM, our method incorporates suboptimal solutions into the loss function, effectively minimizing the average log-likelihood of these suboptimal candidates. This approach improves sample efficiency and accelerates the convergence of model training.

DPO (Rafailov et al., 2023) employs a reference model $\pi_{\text{ref}}$, analogous to RLHF (Stiennon et al., 2020), to regularize the trained model against excessive deviation from the initial policy. The DPO loss function is defined as:

$$\mathcal{L}_{\text{DPO}}(\pi_{\boldsymbol{\theta}}|\pi_{\text{ref}}, \boldsymbol{x}, \boldsymbol{y}_w, \boldsymbol{y}_l) = -\log \sigma \left( \beta \log \frac{\pi_{\boldsymbol{\theta}}(\boldsymbol{y}_w|\boldsymbol{x})}{\pi_{\text{ref}}(\boldsymbol{y}_w|\boldsymbol{x})} - \beta \log \frac{\pi_{\boldsymbol{\theta}}(\boldsymbol{y}_l|\boldsymbol{x})}{\pi_{\text{ref}}(\boldsymbol{y}_l|\boldsymbol{x})} \right), \tag{6}$$

where $\beta$ controls the strength of regularization toward the reference model $\pi_{\text{ref}}$. In contrast, our method eliminates the dependency on an explicit reference model, simplifying the training framework while avoiding potential distributional shift issues.

To simplify the training phase and align the training goal with the generation goal, SimPO (Meng et al., 2024) eliminates the reference model and simplifies the loss function as follows:

$$\mathcal{L}_{SimPO}(\pi_{\boldsymbol{\theta}}|\boldsymbol{x}, \boldsymbol{y}_w, \boldsymbol{y}_l) = -\log \sigma \left( \frac{\beta}{|\boldsymbol{y}_w|} \log \pi_{\boldsymbol{\theta}}(\boldsymbol{y}_w|\boldsymbol{x}) - \frac{\beta}{|\boldsymbol{y}_l|} \log \pi_{\boldsymbol{\theta}}(\boldsymbol{y}_l|\boldsymbol{x}) - \gamma \right), \tag{7}$$

where $\beta$ is a constant that controls the scaling of the difference, and $\gamma$ is a target margin term. In contrast, BOPO uses an adaptive objective gap factor to scale the differences, instead of relying on additional hyperparameters.

Considering the similarity between the BOPO loss and that of policy gradients, we additionally analyze the loss function in the REINFORCE algorithm (Kwon et al., 2020) here:

$$\mathcal{L}_{PG}(\pi_{\boldsymbol{\theta}}|\boldsymbol{x}, \boldsymbol{y}) = -(g(\boldsymbol{y}) - b) \log \pi_{\boldsymbol{\theta}}(\boldsymbol{y}_w|\boldsymbol{x}), \tag{8}$$

where $b$ is a baseline to distinguish positive or negative optimization signals for each sample $\boldsymbol{y}$. In POMO, $b = \sum_i^B g(\boldsymbol{y}_i)/B$ is the average objective of $B$ samples. Note that, both BOPO and REINFORCE use the exact reward (objective) to optimal policy. However, BOPO directly distinguishes optimization signals based on exact preferences, and the strength of the optimization signal correlates with the difference in log-likelihood, requiring the model to maximize the probabilities gap between $\boldsymbol{y}_w$ and $\boldsymbol{y}_l$.

## C.2. Gradient Analysis

Let $z$ denote the argument of the sigmoid in Equation (4):

$$z = \frac{g(\boldsymbol{y}_l)}{g(\boldsymbol{y}_w)} \left( \frac{1}{|\boldsymbol{y}_w|} \log \pi_\theta(\boldsymbol{y}_w|\boldsymbol{x}) - \frac{1}{|\boldsymbol{y}_l|} \log \pi_\theta(\boldsymbol{y}_l|\boldsymbol{x}) \right).$$

The gradient of $\mathcal{L}_{\text{BOPO}}$ with respect to $\theta$ is:

$$\nabla_\theta \mathcal{L}_{\text{BOPO}} = \frac{\partial \mathcal{L}_{\text{BOPO}}}{\partial z} \cdot \nabla_\theta z.$$

Derivative of $-\log \sigma(z)$ becomes:

$$\frac{\partial \mathcal{L}_{\text{BOPO}}}{\partial z} = -(1 - \sigma(z)).$$

Gradient of $z$ with respect to $\theta$:

$$\nabla_\theta z = \frac{g(\boldsymbol{y}_l)}{g(\boldsymbol{y}_w)} \left( \frac{1}{|\boldsymbol{y}_w|} \nabla_\theta \log \pi_\theta(\boldsymbol{y}_w|\boldsymbol{x}) - \frac{1}{|\boldsymbol{y}_l|} \nabla_\theta \log \pi_\theta(\boldsymbol{y}_l|\boldsymbol{x}) \right).$$

Combining these, the total gradient becomes:

$$\nabla_\theta \mathcal{L}_{\text{BOPO}} = \underbrace{-\frac{g(\boldsymbol{y}_l)}{g(\boldsymbol{y}_w)}}_{\text{Adaptive Scaling}} \cdot \underbrace{(1 - \sigma(z))}_{\text{Confidence Weight}} \cdot \left( \underbrace{\frac{1}{|\boldsymbol{y}_l|} \nabla_\theta \log \pi_\theta(\boldsymbol{y}_l|\boldsymbol{x}) - \frac{1}{|\boldsymbol{y}_w|} \nabla_\theta \log \pi_\theta(\boldsymbol{y}_w|\boldsymbol{x})}_{\text{Direction of Policy Update}} \right).$$

**Adaptive Scaling** $\frac{g(\boldsymbol{y}_l)}{g(\boldsymbol{y}_w)}$: For minimization problems, $g(\boldsymbol{y}_w) < g(\boldsymbol{y}_l)$, so $\frac{g(\boldsymbol{y}_l)}{g(\boldsymbol{y}_w)} > 1$. This amplifies the gradient magnitude for pairs where $\boldsymbol{y}_l$ is significantly worse than $\boldsymbol{y}_w$, prioritizing updates that correct large suboptimalities.

**Confidence Weight** $1 - \sigma(z)$: As the policy becomes more confident in preferring $\boldsymbol{y}_w$ over $\boldsymbol{y}_l$ ($\sigma(z) \to 1$), the gradient diminishes. This prevents overfitting to already well-separated pairs.

**Normalization** by $|\boldsymbol{y}|$: The normalization $\frac{1}{|\boldsymbol{y}|}$ ensures that solutions of different lengths contribute equally to the gradient. Without this, longer solutions (e.g., JSP schedules with more operations) would dominate updates.

**Direction of Update**: The gradient increases the likelihood of $\boldsymbol{y}_w$ (since $\nabla_\theta \log \pi_\theta(\boldsymbol{y}_w|\boldsymbol{x})$ is added) and decreases the likelihood of $\boldsymbol{y}_l$ (since $\nabla_\theta \log \pi_\theta(\boldsymbol{y}_l|\boldsymbol{x})$ is subtracted).

Compared with DPO and SimPO, their gradients are as follows:

$$\nabla_\theta \mathcal{L}_{\text{DPO}} = -\beta \cdot \underbrace{(1 - \sigma(d))}_{\text{Confidence Weight}} \cdot \left( \underbrace{\nabla_\theta \log \pi_\theta(\boldsymbol{y}_l|\boldsymbol{x}) - \nabla_\theta \log \pi_\theta(\boldsymbol{y}_w|\boldsymbol{x})}_{\text{Direction of Policy Update}} \right),$$

$$\nabla_\theta \mathcal{L}_{\text{SimPO}} = -\beta \cdot \underbrace{(1 - \sigma(s))}_{\text{Confidence Weight}} \cdot \left( \underbrace{\frac{1}{|\boldsymbol{y}_l|} \nabla_\theta \log \pi_\theta(\boldsymbol{y}_l|\boldsymbol{x}) - \frac{1}{|\boldsymbol{y}_w|} \nabla_\theta \log \pi_\theta(\boldsymbol{y}_w|\boldsymbol{x})}_{\text{Direction of Policy Update}} \right),$$

where

$$d = \beta \left( \log \frac{\pi_{\boldsymbol{\theta}}(\boldsymbol{y}_w|\boldsymbol{x})}{\pi_{\text{ref}}(\boldsymbol{y}_w|\boldsymbol{x})} - \log \frac{\pi_{\boldsymbol{\theta}}(\boldsymbol{y}_l|\boldsymbol{x})}{\pi_{\text{ref}}(\boldsymbol{y}_l|\boldsymbol{x})} \right), \quad s = \beta \left( \frac{1}{|\boldsymbol{y}_w|} \log \frac{\pi_{\boldsymbol{\theta}}(\boldsymbol{y}_w|\boldsymbol{x})}{\pi_{\text{ref}}(\boldsymbol{y}_w|\boldsymbol{x})} - \frac{1}{|\boldsymbol{y}_l|} \log \frac{\pi_{\boldsymbol{\theta}}(\boldsymbol{y}_l|\boldsymbol{x})}{\pi_{\text{ref}}(\boldsymbol{y}_l|\boldsymbol{x})} - \frac{\gamma}{\beta} \right),$$

represent the confidence weight in DPO and SimPO.

DPO and SimPO rely on fixed hyperparameters (i.e., $\beta$ and $\gamma$) to control the gradient magnitude, requiring manual tuning and lacking dynamic adaptability. Furthermore, DPO does not perform length normalization on the log-likelihood calculations, which may lead to instability in scenarios with variable-length outputs. In contrast, BOPO enhances robustness across diverse scenarios through adaptive scaling factors and length normalization, thereby reducing dependence on the $\beta$ hyperparameter.

## D. Runtime Analysis for JSP

We additionally provide the solving time in Table 12, which is an important aspect in some scheduling scenarios.

Our model, MGL, achieves significantly lower solving time compared to all non-constructive methods while delivering performance closing to L2S$_{5k}$. When compared to RL-based greedy constructive methods, MGL maintains competitive solving time even when sampling 512 solutions. Notably, MGL's solving time does not increase significantly with larger $B'$, a clear distinction from GAT-MHA, the model proposed in SLIM. This is because GAT-MHA relies on MHA decoder, which has a computational complexity of $O(n^2)$, a drawback that becomes particularly pronounced as $B'$ grows.

## E. Further Analysis of Hybrid Rollout

To investigate the impact of introducing a greedy solution through hybrid rollout, we trained the model using both hybrid rollout and pure sampling approaches under different numbers of generated solutions $B$. The comparison results are shown

*Table 12.* The average solving time (s) of algorithms on the TA benchmark.

| Shape | Non-constructive | | | | Greedy Constructive | | | | $B'$=128 | | $B'$=512 | |
| | OR-Tools | L2S$_{500}$ | TGA$_{500}$ | L2S$_{5k}$ | PDRs | L2D | SchN | CL | SLIM | BOPO | SLIM | BOPO |
|---|---|---|---|---|---|---|---|---|---|---|---|---|
| 15x15 | 462 | 9.3 | 12.6 | 92.2 | 0.00 | 0.39 | 3.5 | 0.80 | 0.69 | 0.67 | 0.72 | 0.75 |
| 20x15 | 2880 | 10.1 | 14.6 | 102 | 0.00 | 0.60 | 6.6 | 1.10 | 0.84 | 0.80 | 1.07 | 0.96 |
| 20x20 | 3600 | 10.9 | 17.5 | 114 | 0.00 | 0.72 | 11 | 1.39 | 1.11 | 1.06 | 1.37 | 1.23 |
| 30x15 | 3600 | 12.7 | 17.2 | 120 | 0.01 | 0.95 | 17.1 | 1.49 | 1.24 | 1.19 | 1.83 | 1.44 |
| 30x20 | 3600 | 14 | 19.3 | 144 | 0.01 | 1.41 | 28.3 | 1.72 | 1.66 | 1.59 | 2.42 | 1.91 |
| 50x15 | 3600 | 16.2 | 23.9 | 168 | 0.01 | 1.81 | 52.5 | 2.82 | 2.19 | 1.99 | 4.06 | 2.60 |
| 50x20 | 3600 | 22.8 | 24.4 | 228 | 0.02 | 3.00 | 96 | 3.93 | 2.91 | 2.63 | 5.41 | 3.44 |
| 100x20 | 3600 | 50.2 | 42.0 | 504 | 0.19 | 9.39 | 444 | 9.58 | 7.85 | 5.31 | 20.05 | 7.96 |

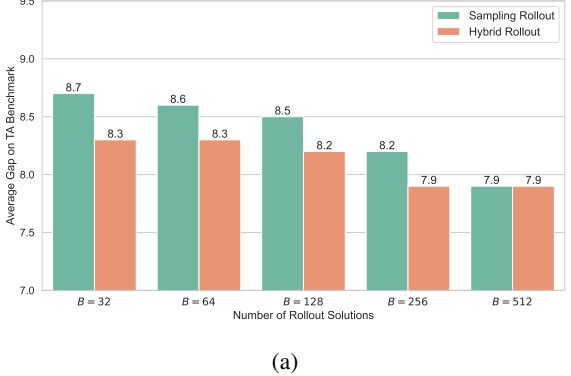

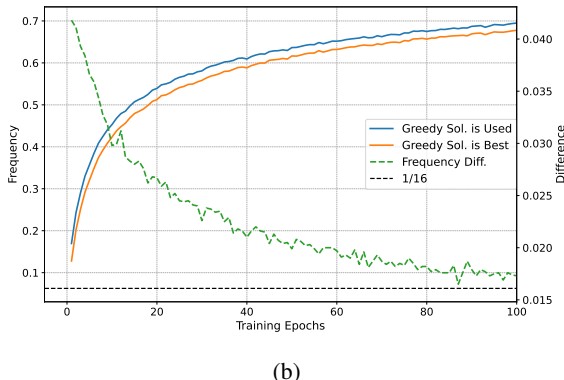

(a)  (b)

*Figure 5.* Analysis of Hybrid Rollout. (a) Average gap (%) on TA benchmark of different rollout methods with varying numbers of generated solutions $B$, (b) Participation and optimality of greedy solution during training.

in Figure 5a. Evidently, hybrid rollout not only improves model performance but also significantly reduces the dependency on $B$. Notably, the model achieves competitive results even with smaller generated sizes $B$. For instance, $B = 32$ with hybrid rollout performs similarly to $B = 64$ with hybrid rollout and is only 0.4 points behind $B = 512$ with hybrid rollout. In contrast, $B = 32$ without hybrid rollout performs poorly, with a gap of 0.8 compared to $B = 512$ without hybrid rollout. Notably, hybrid rollout enables a generated solutions size of 256 to match the performance of a generated solutions size of 512 without hybrid rollout, demonstrating equivalent efficacy with reduced computational demand.

As illustrated in Figure 5b, to investigate how greedy solutions enhance training efficacy, we trained the model on TSP20 with hyperparameters $B = 128$ and $K = 8$, systematically tracking two metrics per epoch:

- The frequency of greedy solutions selected for preference pair construction (theoretical random selection probability: $1/16$);

- The frequency of greedy solutions being identified as the best solution.

The rising selection frequency of greedy solutions demonstrates their growing dominance in training. Given that greedy solutions exhibit the highest log-likelihood among all candidates, their participation delivers stronger gradient signals, accelerating model convergence. The synchronized increase in greedy solutions being recognized as the best solution, and the decrease of difference between the frequencies mentioned above validates their critical role in improving solution set quality. This inherently addresses the limitation of sampling rollout, where suboptimal exploration often fails to capture high-quality candidates.

These demonstrate that introducing a greedy solution via hybrid rollout has an effect akin to increasing the solution generated size $B$ during training, but without incurring additional computational costs. This is particularly significant because large-scale and complex COPs often require substantial resources. Hybrid rollout offers a memory-efficient alternative while maintaining high performance.

## F. Additional Experiments.

To further investigate the impact of $B$ on different problem types, we evaluate our method on TSP20/50 with $B = 20/50$ (denoted as BOPO$^-$). As shown in Table 13, when sampling the same number of solutions as POMO, the performance of the model remains comparable to $B = 128$. This suggests that for small-scale or simple problems, the optimal $B$ is low, as the model can efficiently sample high-quality solutions with fewer times.

BOPO's advantage primarily stems from pairwise preference learning, rather than simply being an RL variant with the proposed filtering method. To demonstrate this point, we compare BOPO with a POMO variant employing our Hybrid Rollout and Uniform Filtering (denoted as POMO$^+$). The key difference lies in the loss: while BOPO uses pairwise loss, POMO$^+$ calculates a standard POMO loss based on the filtered $K$ solutions, as it lacks the pairwise mechanism. Results in Table 13 show that POMO$^+$ is inferior to BOPO, demonstrating the effectiveness of the preference learning of BOPO. Moreover, POMO$^+$ is even inferior to POMO, underscoring that the filtering method is tightly adapted to BOPO and may not be applicable to general RL methods.

*Table 13.* Average gaps (%) on generated TSP instances.

| Methods | TSP20 | TSP50 |
|---------|-------|-------|
| POMO | 0.002 | 0.042 |
| POMO$^+$ | 0.005 | 0.081 |
| BOPO$^-$ | **0.000** | **0.009** |
| BOPO | **0.000** | **0.009** |

