# OpenReview forum: "BOPO: Neural Combinatorial Optimization via Best-anchored and Objective-guided Preference Optimization"
_ICML.cc/2025/Conference — ICML 2025 poster_

### Official Review · Reviewer_Madu · 2025-02-27

**Overall Recommendation:** 3

**Summary:**

The paper proposes to use (a specific variant of) preference-based RL losses instead of regular reward-based policy gradients for combinatorial optimization problems.

The primary finding is that this Preference Optimization for Combinatorial Optimization (POCO) loss improves sample complexity and asymptotic performance over standard (PPO) and specialized baselines (SLIM), over multiple different combinatorial optimization problems.

This is a surprising result, because the original reason preference losses are used is due to lack of precise rewards (e.g. with human feedback). I would've thought that for combinatorial optimization which indeed has precise rewards, using preferences (which is imprecise) can be worse.

**Claims And Evidence:**

Because of how unintuitive the results are (i.e. even with precise rewards, preference-based loss leads to better outcomes), I'm inclined to think of possible subtle numeric effects which led to this result. For example:
  * Reward normalization can be tricky with exact rewards, and training the value-head to predict advantages typically can cause problems if not careful.
  * For $n$ objective evaluations, preference optimization uses ${n \choose 2}$ pairs, while policy gradient would only use $n$, so preference optimization performs more gradient updates due to more data. The paper proposes filtering methods (uniform filtering and best-anchored pairing) to reduce the data size and only focus on diverse or high quality solutions.
    * If this is crucial to improving performance, I'm wondering if there would've also been a policy gradient variant (e.g. add a numeric transformation to upweight large rewards) that would've equally done well.

I'm not asking the authors to ablate these (since this could require a lot of work), but I'm just stating that the devil's in the details.

**Essential References Not Discussed:**

N/A

**Experimental Designs Or Analyses:**

Please see my "Claims and Evidence" part. I'm sure that the authors did a solid job on running multiple experiments and baselines, but experimental RL requires a lot of small little subtle details which can strongly affect results.

**Methods And Evaluation Criteria:**

Yes, the paper evaluates over multiple different combinatorial optimization problems (Job-shop scheduling problem, Traveling Salesman Problem, Flexible Job-shop scheduling problem), with multiple baselines.

I do think the presentation could be improved however (too many abbreviations and large tables, and too many random details in the text). It's possible to improve the experimental presentation pictorially using bars or graphs.

Overall it's not a priority IMO to show outperformance over very domain-specific combinatorial optimization baselines - what is a priority, like I mentioned earlier, is the conclusion that preference-based RL beats exact-reward RL.

**Other Comments Or Suggestions:**

I'm not particularly for or against this paper - i.e. I'm pretty lukewarm about it, because:

1. When it comes to the actual combinatorial optimization improvements, the gains are fairly incremental, and overall combinatorial optimization has become very saturated.

2. The result that preference-based optimization outperforms exact rewards, _is_ interesting, but the paper isn't investigating this fundamental result in-so-much-as applying it for combinatorial optimization

**Other Strengths And Weaknesses:**

# Strengths
Using RL-HF techniques like DPO for exact-reward problems, outside of human feedback, is an important topic for linking the field of LLM alignment with traditional RL, and I applaud this paper in doing so.

# Weaknesses
The current presentation of the paper makes the contribution incremental and not fundamental - i.e. due to lots of random details and abbreviations stuffed into the writing, the overall read can look like "We tried a different tweak and got slightly better results in a very specific domain". I would strongly encourage the authors to avoid this style of writing.

**Questions For Authors:**

See my concerns above.

**Relation To Broader Scientific Literature:**

It is interesting to see how some of the RL-HF literature (preference-based learning) can be adopted to classic RL (e.g. combinatorial optimization). Originally, preference-based losses were only created to deal with the noisiness of human feedback, but I would've never expected it to do better than traditional policy gradients for exact rewards.

**Theoretical Claims:**

Not applicable, no theory.

---

> ### Author Rebuttal · Authors · 2025-03-31
>
> Thank you for your valuable comments.
>
> >**Q1:** I would've thought that for combinatorial optimization which indeed has precise rewards, using preferences (which is imprecise) can be worse.
>
> We agree that precise rewards are also crucial to combinatorial optimization. **Meanwhile, we would like to clarify that POCO, unlike standard preference optimization techniques used in LLM, incorporates the precise objective value (i.e., reward in RL) into our loss function.** As highlighted in our submission, this key contribution stems from our insight into the critical role of precise rewards in combinatorial optimization.
>
> Specifically, the proposed POCO loss is $\log\sigma\left(\frac{g(y_l)}{g(y_w)}\left(\frac{\log\pi_\theta(y_w|x)}{|y_w|}-\frac{\log\pi_\theta(y_l|x)}{|y_l|}\right)\right)$, where $\frac{g(y_l)}{g(y_w)}$ is a scaling factor derived from precise objective values for the best solution and the inferior one.
>
> The results in Table 6 and discussions in Section 5.4 demonstrate the effectiveness of incorporating precise objective values into the POCO loss, e.g., POCO reduces the gap to 12.9% on the DMU benchmark, outperforming its variant without this scaling factor (gap 13.2%).
>
>
> >**Q2:** I'm inclined to think of possible subtle numeric effects which led to this result.
>
> While we acknowledge that numerical artifacts can sometimes confound comparisons, we emphasize that our core contribution, a novel preference-based training framework, is not attributable solely to such effects.
>
> * **By leveraging preference learning between the best solution and diverse inferior ones, POCO guides the model toward promising decision trajectories and discern suboptimal choices, demonstrating its clear advantage against RL methods.** POCO exhibits significantly superior solution quality and higher training efficiency ($48\times$ speedup for JSP), as shown in the response to Reviewer upvg Q5 and Q1, respectively. Note that POMO, a REINFORCE-based method, uses terminal rewards and computes baselines via means, eliminating value-head estimation and minimizing numeric effect risks.
>     * Here is a closer look at policy gradient. POMO uses a baseline to distinguish positive/negative optimization signals for each sample:
> $(g(y) - b) \nabla_\theta \log \pi_{\theta}(y|x).$
> In contrast, POCO directly distinguishes based on exact preferences:
> $\frac{g(y_l)}{g(y_w)}\frac{(1-\sigma(z))}{|y_w|}\nabla_\theta\log\pi_\theta(y_w|x)-\frac{g(y_l)}{g(y_w)}\frac{(1-\sigma(z))}{|y_l|}\nabla_\theta \log\pi_\theta(y_l|x),$
> where $z=\frac{g(y_l)}{g(y_w)}\left(\frac{1}{|y_w|}\log \pi_\theta(y_w|x)-\frac{1}{|y_l|}\log\pi_\theta(y_l|x)\right)$. The optimization signal strength correlates with the log-likelihood difference $z$, requiring the model to maximize the probabilities gap between $y_w$ and $y_l$. This results in faster convergence and better performance.
> * **POCO's advantage primarily stems from pairwise preference learning, rather than simply being a RL variant with the proposed filtering method.** To demonstrate this point, we compare POCO with a POMO variant employing our Hybrid Rollout and Uniform Filtering (denoted as POMO+). The key difference lies in the loss: while POCO uses pairwise loss, POMO+ calculates a standard POMO loss based on the filtered $K$ solutions, as it lacks the pairwise mechanism. Results in Table H show that POMO+ is inferior to POCO, demonstrating the effectiveness of the preference learning of POCO. Moreover, POMO+ is even inferior to POMO, underscoring that the filtering method is tightly adapted to POCO and may not be applicable to general RL methods.
>
> Based on the above two points, we would like to reiterate that POCO's unique advantages stem from its preference learning and the two tightly coupled designs (the pair construction method and the loss function based on objective values), rather than simply relying on subtle numeric effects.
>
> **Table H: Gaps (%) on TSP**
> Method|TSP20|TSP50
> -|-|-
> POMO|0.002|0.042
> POMO+|0.005|0.081
> POCO|**0.000**|**0.009**
>
> >**Q3:** Mainly Section C, to verify what types of RL losses are being used.
>
> We have stated typical RL losses in Section 3.1, including REINFORCE for routing problems and PPO for scheduling problems. According to your suggestions, we will add a detailed description and analysis of these RL losses to Appendix C.
>
> >**Q4:** The current presentation of the paper makes the contribution incremental and not fundamental ...
>
> Our main contribution is a novel training framework, distinctly different from traditional RL, that not only improves performance but also convergence speed. We will thoroughly proofread the paper to highlight this core contribution.
>
> >**Q5:** ..., the gains are fairly incremental, and overall combinatorial optimization has become very saturated.
>
> In the NCO field, such improvements are considered significant. POCO delivers substantial performance gains and faster convergence, as demonstrated in our responses to Reviewer upvg Q5 and Q1, respectively.

---

> > ### Comment · Reviewer_Madu · 2025-04-01
> >
> > Thank you for the clarification, especially regarding the RL loss still taking into account the objective values (and not just pairwise comparison). I will increase my score, although I am still lukewarm about this paper.

---

> > > ### Author Response · Authors · 2025-04-02
> > >
> > > Thank you for your clarification and for acknowledging our explanation. We appreciate your recognition and are encouraged by the increased score, even if you remain cautiously optimistic about the paper.

---

### Official Review · Reviewer_jKth · 2025-02-28

**Overall Recommendation:** 4

**Summary:**

This paper first introduce the concept of Preference Optimization to the area of NCO. Generally, as the expected advantage of PO, POCO demonstrates instance-efficiency comparing to RL and SLL. (As shown in Figure 2 and Figure 3)

## update after rebuttal:

This is a novel and inspiring paper, so I keep my positive rating.

**Claims And Evidence:**

Some claims in the paragraph ``Comparison with Other Losses.`` are not clear. The reference model is actually a constraint for LLM, I do not agree that this can be a characteristic of the POCO loss. Instead, the Adaptive Scaling part can distinguish DPO and the POCO loss. POCO loss excludes the target reward margin term in SimPO but this is not discussed in that paragraph.

**Essential References Not Discussed:**

This paper has included prior related findings.

**Experimental Designs Or Analyses:**

I recommend to also includes SL-based POMO as baseline. [1]

[1] Yao S, Lin X, Wang J, et al. Rethinking Supervised Learning based Neural Combinatorial Optimization for Routing Problem[J]. ACM Transactions on Evolutionary Learning, 2024.

**Methods And Evaluation Criteria:**

Generally okay.

**Other Comments Or Suggestions:**

Please refer to Questions For Authors.

**Other Strengths And Weaknesses:**

Strength:

1. Introducing Preference Optimization to NCO is novel.

2. Compared to RL-based and SLL-based baselines, POCO exhibits instance-efficiency in training.
﻿
Weakness:

Generally, there are no major weaknesses in this paper. One minor weakness of this paper is that the preparation stage for POCO (i.e., ``Preference Pair Construction``) can be time-consuming compared to RL-based methods. So POCO can hardly be extended to train on large-scale CO.

**Questions For Authors:**

1. Section 3.1. Neural Combinatorial Optimization (NCO) provides a detailed introduction to RL methods. As far as I know, there is not much work using PPO in NCO training, so is this part of the description a bit redundant?

2. The design of Adaptive Scaling is reasonable, but my concern is that the distribution of objective function values is often different over different CO problems. Will using raw scaling factors lose stability on different problems? Can rank-based factors or factors after normalization be better?

3. Can you provide a rough preparation time (used for ``Preference Pair Construction``) for each instance?

4. Can POCO generalize to NCO models other than POMO on TSP? It would be great if POCO could still demonstrate its performance on other models.

**Relation To Broader Scientific Literature:**

This paper discuss a new training paradigm for NCO, i.e., Preference Optimization. POCO shows advantages in intance-efficiency and final performance.

**Theoretical Claims:**

This paper do not contain theoretical claims.

---

> ### Author Rebuttal · Authors · 2025-03-31
>
> Thank you for your valuable comments.
>
> >**Q1:** I recommend to also includes SL-based POMO as baseline.
>
> We have added results from the SL-based method [1] to the table E. Since all methods use the same model, differences stem from training algorithms.
>
> **POCO achieves comparable or better performance than SL-based methods without requiring labeled optimal solutions, further demonstrating the superiority of POCO**. Results show that on TSP20, SL-based and POCO perform identically. On TSP50, SL-based slightly outperforms POCO (0.001% vs 0.01%). However, on TSP100, POCO is superior (0.04% vs 0.05%).
>
> **Table E: Gaps (%) on TSP Instances**
> Method|TSP20|TSP50|TSP100
> -|-|-|-
> POMO(aug)|0.00|0.03|0.14
> SLIM(aug)|0.01|0.15|1.17
> SL(aug)|**0.00**|**0.00**|0.05
> POCO(aug)|**0.00**|0.01|**0.04**
>
> [1] Rethinking Supervised Learning based Neural Combinatorial Optimization for Routing Problem. ACM Transactions on Evolutionary Learning, 2024.
>
> >**Q2:** the preparation stage for POCO can be time-consuming compared to RL-based methods. Can you provide a preparation time for each instance?
>
> This problem is mainly caused by us sampling more solutions (larger $B$). We provide the following clarification:
>
> 1. **POCO does not necessarily require sampling a large number of solutions to maintain performance**. As noted in our response to Reviewer upvg Q4, POCO still achieved superior performance when sampling the same number of solutions as POMO, although we recommend an best value of 256 for $B$.
> 2. **Training time does not significantly increase with larger** $B$. Table F shows the approximate training time per epoch for different problem scales and $B$. For small-scale problems, the training time has hardly changed. For larger problems like TSP100, even if $B$ is increased by 2.5 times, the training time only increases by 1.6 times.
>
> **Table F: Training Time Per Epoch**
> $B$|TSP20|TSP50|TSP100
> -|-|-|-|
> 20/50/100|2m30s|4m30s|12m
> 128|2m30s|5m|-
> 256|-|-|20m
>
> >**Q3:** As far as I know, there is not much work using PPO in NCO training, so is this part of the description a bit redundant?
>
> We acknowledge that REINFORCE is the dominant training algorithm for routing problems. However, as noted in our paper, PPO is a typical method for scheduling problems [2-4]. We remain open to further refinements to enhance clarity and relevance.
>
> [2] Learning to dispatch for job shop scheduling via deep reinforcement learning. NeurIPS 2020.
>
> [3] Flexible job-shop scheduling via graph neural network and deep reinforcement learning. IEEE Transactions on Industrial Informatics, 2023.
>
> [4] Solving flexible job shop scheduling problems via deep reinforcement learning. Expert Systems with Applications, 2024.
>
> >**Q4:** Will using raw scaling factors lose stability on different problems? Can rank-based factors or factors after normalization be better?
>
> This is a very interesting question. We agree this is a valuable direction for future research.
>
> 1. We observed that scaling factors vary across different problems, with distributions differing significantly. For some challenging problems, such as those where objective values have large or minimal gaps, raw scaling factors might fail. This is an important area for further exploration.
> 2. Rank-based factors offer stability in any scenario but may not accurately reflect true solution differences, potentially leading to suboptimal performance. Normalized factors, however, seem promising, with techniques like normalization, clipping, or compression functions offering potential improvements.
>
> >**Q5:** Can POCO generalize to NCO models other than POMO on TSP?
>
> We would like to emphasize that POCO, like POMO, is a general and fundamental training algorithm. POMO serves as the backbone for most SOTA methods, so naturally, the backbone of these SOTA methods can also be replaced with the superior POCO. To further showcase this potential, we conducted additional experiments on InViT [5].
>
> Specifically, we adopt POCO to train InViT, but due to memory constraints, we set the batch size to 64 for InViT (also $B$ for InViT+POCO), with $K=8$. We retrained InViT with identical parameters. Results in Table G shows that InViT+POCO outperformes InViT on various distributions and larger-scale problems.
>
> **Table G: Gaps (%) on TSP**
> Method|U-1k|U-5k|U-10k|C-1k|C-5k|C-10k
> -|-|-|-|-|-|-
> InViT-2V|6.49|8.21|6.61|10.01|10.36|9.23
> InViT-2V+POCO|**5.98**|**7.67**|**6.00**|**9.58**|**9.81**|**9.18**
>
> Method|E-1k|E-5k|E-10k|I-1k|I-5k|I-10k
> -|-|-|-|-|-|-
> InViT-2V|9.61|11.58|9.72|7.95|9.42|7.39
> InViT-2V+POCO|**8.67**|**10.59**|**8.94**|**7.38**|**8.88**|**7.04**
>
> Distribution: Uniform (U), Cluster (C), Explosion (E), Implosion (I). The numbers behind represent the problem scales. Note that we report the results both for 300 epochs due to extended training time. Full results for 800 epochs will be provided in the subsequent discussion phase.
>
> [5] InViT: A generalizable routing problem solver with invariant nested view transformer. ICML 2024.

---

> > ### Comment · Reviewer_jKth · 2025-04-05
> >
> > Thanks for your response. I will keep my accept score.

---

> > > ### Author Response · Authors · 2025-04-06
> > >
> > > Thank you for your feedback and for maintaining your accept score. We greatly appreciate your support and consideration.

---

### Official Review · Reviewer_xAAW · 2025-03-12

**Overall Recommendation:** 3

**Summary:**

The introduction of POCO, a training paradigm for NCO to enhance sample efficiency. This is accomplished by  (1) design a preference pair construction method for improving exploration and exploitation, (2) gradual building of the loss function. This is evaluated on three problems Job shop scheduling, TSP, and FJTSP.

## Update after rebuttal:
The authors have addressed most of my comments. I would like to thank them for the efforts. I will increase my score to 3.

**Claims And Evidence:**

Yes.

**Essential References Not Discussed:**

N/A

**Experimental Designs Or Analyses:**

Yes. They are mostly sound.

**Methods And Evaluation Criteria:**

Yes.

**Other Comments Or Suggestions:**

See Other Strengths And Weaknesses.

**Other Strengths And Weaknesses:**

## Strengthens:
- No need for labels for the training graphs.
- OOD evaluation at inference.
- POCO evaluation with different backbone models.

## Weaknesses:

-	Subpar results when compared to SOTA heuristic Concorde. Results are only good when compared to other NCO methods. Concorde is not used for TSP. Is there a reason? In general, what is the advantage of using this NCO when compared to LKH3 and Concorde?

-	How about TSP-1000?

-	Scalability results are missing for TSP? can the authors test their trained model on instances where the ILP fails?

**Questions For Authors:**

See Other Strengths And Weaknesses.

**Relation To Broader Scientific Literature:**

The authors provide a new training paradigm for NCO that is architecture-agnostic.

**Theoretical Claims:**

N/A

---

> ### Author Rebuttal · Authors · 2025-03-31
>
> We thank the reviewer for the encouraging feedback and incisive questions.
>
> >**Q1:** Concorde is not used for TSP.
>
> We have included Concorde's results from POMO [1] and added them to the table C for comparison. Concorde, LKH3, and Gurobi are traditional iterative search-based algorithms with identical performance (0.0% gap), differing only in solving time.
>
> LKH3, being a strong heuristic solver, can offer fast approximate solutions for large-scale problems and even delivery the optimal solutions for small-scale problems. Including Concorde's results does not alter our existing conclusions, as we have already incorporated other strong heuristic methods.
>
> **Table C: Gaps (%) on TSP Instances**
>
> |Method|TSP20 Gap|TSP20 Time|TSP50 Gap|TSP50 Time|TSP100 Gap| TSP100 Time|
> |-|-|-|-|-|-|-|
> |Concorde|0.00|5m|0.00|13m|0.00|1h|
> |Gurobi|0.00|42s|0.00|6m|0.00|25m|
> |LKH3|0.00|7s|0.00| 2m|0.00|17m|
> |POMO (aug.)|0.00|3.6s|0.03|6.6s|0.14| 18.1s|
> |SLIM (aug.)|0.01| 3.6s|0.15|6.6s|1.17|18.1s|
> |POCO (aug.)|**0.00**|3.6s|**0.01**|6.6s|**0.04**|18.1s|
>
> [1] POMO: Policy optimization with multiple optima for reinforcement learning. NeurIPS 2020.
>
> >**Q2:** Subpar results when compared to SOTA heuristic Concorde. In general, what is the advantage of using this NCO when compared to LKH3 and Concorde?
>
> Unlike traditional iterative search-based methods, NCO without expert knowledge balances solving time and performance, so its contribution should not be evaluated solely based on performance.
>
> * **A key advantage of NCO is its ability to achieve near-optimal solutions with significantly shorter solving times, especially for large-scale problems**：
>     * For TSP100, POCO achieves a near-optimal gap of 0.04% while reducing solving time dramatically (17m for LKH3 vs. 18s for POCO).
>     * On large-scale JSP problems (TA 100x20), NCO demonstrates clear advantages. Exact algorithms like Gurobi and OR-Tools, within 1 hour (see Table 12 in Appendix D), achieve 11% and 3.9% gap (see Table 1), respectively, which are inferior to the 1.4% gap POCO achieves in just 8 seconds (see Table 12 and Table 1, respectively).
> * **Another advantage of NCO is that it does not require expert knowledge to design complex algorithms**, thereby expanding its applicability to a wider range of scenarios.
>
> >**Q3:** How about TSP-1000? Can the authors test their trained model on instances where the ILP fails?
>
> To further showcase the potential of POCO on large-scale problems, we conducted additional experiments on InViT [2] framework, which is designed to solve large-scale problems where the ILP fails. Note that POMO is not designed for large-scale problems, but we have also demonstrated the superiority of POCO with generalization experiments on TSPLIB (see Table 3).
>
> Main results generalizing to various distributions and larger-scales are shown in Table D. InViT+POCO outperformes InViT on all distributions and scales, underscoring its superiority. At these scales, Gurobi and LKH3 require 30 minutes to 1.5 days to solve, whereas NCO needs only up to approximately 10 minutes.
>
> **Table D: Gaps (%) Across Different Distributions and Scales**
>
> |Method|U-1k|U-5k|U-10k|C-1k|C-5k|C-10k|
> |-|-|-|-|-|-|-|
> |InViT-2V|6.49|8.21|6.61|10.01|10.36|9.23|
> |InViT-2V+POCO|**5.98**|**7.67**|**6.00**|**9.58**|**9.81**|**9.18**|
>
> |Method|E-1k|E-5k|E-10k|I-1k|I-5k|I-10k|
> |-|-|-|-|-|-|-|
> |InViT-2V|9.61|11.58|9.72|7.95|9.42|7.39|
> |InViT-2V+POCO|**8.67**|**10.59**|**8.94**|**7.38**|**8.88**|**7.04**|
>
> "U": Uniform, "C": Cluster, "E": Explosion, "I": Implosion distributions. The numbers behind represent the problem scales. Note that we report the results both for 300 epochs due to extended training time. Full results for 800 epochs will be provided in the subsequent discussion phase.
>
> [2] InViT: A generalizable routing problem solver with invariant nested view transformer. ICML 2024.

---

### Official Review · Reviewer_upvg · 2025-03-14

**Overall Recommendation:** 3

**Summary:**

This paper proposes POCO (Preference Optimization for Combinatorial Optimization), a method that integrates preference learning into reinforcement learning (RL) to address combinatorial optimization problems. Additionally, the authors introduce Hybrid Rollout, Uniform Filtering, and Best-anchored Pairing as methods for constructing preference pairs used in preference learning. The proposed approach is evaluated and tested on combinatorial optimization tasks including the Job-shop Scheduling Problem (JSP), Traveling Salesman Problem (TSP), and Flexible Job-shop Scheduling Problem (FJSP).

**Claims And Evidence:**

This paper proposes a method for training combinatorial optimization (CO) solver models using preference learning and introduces specific methods for constructing preference pairs required for this approach. However, the rationale behind applying preference learning to train CO solver models is insufficiently explained. Although the introduction states that preference learning was introduced to improve sample efficiency, it remains unclear precisely how sample efficiency is enhanced by this method. Moreover, similar to most existing Neural CO studies, all three experiments performed in this paper utilized *generated* training datasets.

Experiments were conducted on Job-shop Scheduling Problem (JSP), Traveling Salesman Problem (TSP), and Flexible Job-shop Scheduling Problem (FJSP). However, the experimental setup for TSP appears inadequate. Further details regarding this issue are provided in the 'Experimental Designs Or Analyses' section.

**Essential References Not Discussed:**

Luo, Fu, et al. "Neural combinatorial optimization with heavy decoder: Toward large scale generalization." NeurIPS 2023.
Jin, Yan, et al. "Pointerformer: Deep reinforced multi-pointer transformer for the traveling salesman problem." AAAI 2023.
Fang, Han, et al. "Invit: A generalizable routing problem solver with invariant nested view transformer." ICML 2024.

**Experimental Designs Or Analyses:**

Regarding the experiments conducted on TSP instances, there exist several recent studies (e.g., LEHD, Pointerformer, InViT) that have improved upon POMO (Kwon et al., 2020) for solving the TSP. Although SLIM is a recently proposed method included in this paper's comparisons, it performs worse than POMO in most experimental settings presented in Tables 2 and 3. Therefore, to better demonstrate the effectiveness of the proposed method, it would be beneficial to compare it against these recent state-of-the-art studies that have shown superior performance over POMO. Additionally, in the uniformly generated TSP instance experiments presented in Table 2, the node sizes used are relatively small compared to those evaluated in recent studies. It is recommended to perform comparisons on larger-scale instances.

In Figure 3, POMO generates 20 solutions per instance, whereas POCO generates 128 solutions per instance. Due to this discrepancy in the number of solutions generated by each method per instance, it is difficult to consider this experiment as a fair comparison. Furthermore, the performance gap between POMO and POCO reported in Figure 3 is only 0.0013, which is too small to be considered a meaningful difference.


Luo, Fu, et al. "Neural combinatorial optimization with heavy decoder: Toward large scale generalization." NeurIPS 2023.
Jin, Yan, et al. "Pointerformer: Deep reinforced multi-pointer transformer for the traveling salesman problem." AAAI 2023.
Fang, Han, et al. "Invit: A generalizable routing problem solver with invariant nested view transformer." ICML 2024.

**Methods And Evaluation Criteria:**

As mentioned earlier, the benefits of applying preference learning proposed in this paper are not clearly demonstrated.

Furthermore, in the preference pair construction method, rollouts are conducted B times, but only B/K of these rollouts are utilized for model updates, discarding the remaining data. This approach inevitably results in a loss of training resources due to discarding rollout data by a factor of K. Given the same amount of time and computational resources, it remains unclear what specific advantages can be achieved by increasing the values of B and K.

**Other Comments Or Suggestions:**

None

**Other Strengths And Weaknesses:**

None

**Questions For Authors:**

None

**Relation To Broader Scientific Literature:**

None

**Theoretical Claims:**

N/A

---

> ### Author Rebuttal · Authors · 2025-03-31
>
> Thank you for your valuable comments.
>
> >**Q1:** it remains unclear precisely how sample efficiency is enhanced.
>
> We have analyzed POCO's higher sample efficiency in Section 5.4. We further precisely demonstrate significant improvements in sample efficiency by supplementing quantitative results. Note that we retrained models on TSP20/50 with $B=20/50$ & $K=8$, ensuring a fair comparison.
>
> POCO achieves faster convergence than RL when using the same numbers of samples and gradient steps:
>
> 1. POCO achieves approximately $48\times$, $20\times$, and $3\times$ higher training efficiency than RL methods on JSP, TSP20, and TSP50, respectively. Specifically, POCO requires only 2500 training steps, 10 epochs, and 70 epochs to match the gaps achieved by RL methods, which take 120000 steps, 200 epochs, and 200 epochs, respectively.
> 2. POCO achieves significantly superior solution quality within the same number of training epochs. On JSP, POCO attains 0.76%/4.12% lower gaps than RL/SLIM in just 5 epochs. For TSP20/50, POCO achieves 0.0076%/0.0687% lower gaps than POMO in 100 epochs.
>
> >**Q2:** it remains unclear what specific advantages can be achieved by increasing the values of $B$ and $K$.
>
> The advantages of increasing $B$ and $K$ under the same time and computational resources can be seen in Section 5.5.
>
> 1. Increasing $B$ improves performance while increasing memory usage. As shown in Figure 4(a), the gap reduces from 8.3% to 7.9% as $B$ increases from 32 to 512, plateauing at $B=256$. Beyond this point, increasing $B$ provides negligible improvement, as sampling more from $\pi$ rarely yields better solutions. This result demostrates our reasonable setting of $B=256$.
> 2. As presented in Figures 4(b-c), the recommended value for $K$ is 16, and it should scale proportionally with $B$.
>
> >**Q3:** comparison with recent SOTA studies.
>
> We would like to emphasize that POCO, like POMO, is a general and fundamental training algorithm. POMO serves as the backbone for most SOTA methods, so naturally, the backbone of these SOTA methods can also be replaced with the superior POCO. To further showcase this potential, we conducted additional experiments on InViT, which has demonstrated superior performance over LEHD and Pointerformer in its paper.
>
> Specifically, we adopt POCO to train InViT, but due to memory constraints, we set the batch size to 64 for InViT (also $B$ for InViT+POCO), with $K=8$. We retrained InViT with identical parameters. Results in Table A shows that InViT+POCO outperformes InViT on various distributions and larger-scale problems.
>
> **Table A: Gaps (%) on TSP**
> Method|U-1k|U-5k|U-10k|C-1k|C-5k|C-10k
> -|-|-|-|-|-|-
> InViT-2V|6.49|8.21|6.61|10.01|10.36|9.23
> InViT-2V+POCO|**5.98**|**7.67**|**6.00**|**9.58**|**9.81**|**9.18**
>
> Method|E-1k|E-5k|E-10k|I-1k|I-5k|I-10k
> -|-|-|-|-|-|-
> InViT-2V|9.61|11.58|9.72|7.95|9.42|7.39
> InViT-2V+POCO|**8.67**|**10.59**|**8.94**|**7.38**|**8.88**|**7.04**
>
> Distribution: Uniform (U), Cluster (C), Explosion (E), Implosion (I). The numbers behind represent the problem scales. Note that we report the results both for 300 epochs due to extended training time. Full results for 800 epochs will be provided in the subsequent discussion phase.
>
> >**Q4:** Due to this discrepancy in the number of solutions generated by each method per instance, it is difficult to consider this experiment as a fair comparison.
>
> For a fair comparison, we retrained models on TSP20/50 with $B=20/50$ & $K=8$, ensuring it samples the same number of solutions during training as POMO.
>
> Despite reducing $B$, POCO maintains performance comparable to $B=128$, with superior solution quality (see Table B), higher training efficiency (see response to Q1), and better generalization (see response to Q3).
>
> **Table B: Gaps (%) on TSP.**
> Method|TSP20|TSP50
> -|-|-
> POMO (aug)|0.0016|0.0418
> POCO (aug)|**0.0000**|**0.0085**
>
> >**Q5:** the performance gap is too small to be considered a meaningful difference.
>
> 1. For well-optimized small-scale or easy problems, such as TSP20, achieving further improvement, e.g., 0.0013% gap to the optimal solution, is quite challenging. However, the improvements become more pronounced for larger-scale or harder problems: (1) On larger-scale TSP100, POCO's gap 0.04% vs POMO's 0.14% (see Table 2); (2) On the harder JSP, POCO's gap 12.9% vs $SLIM_{MGL}$'s 16.5% on DMU benchmark (see Table 1); (3) On much larger-scale unseen TSPLIB instances with 500-1000 nodes, POCO's gap 22.44% vs POMO's 30.14% (see Table 3).
> 2. In the NCO field, such improvements are considered significant, as evidenced by literature from top AI conferences [1,2]. In [1], among 12 TSP cases, 10 cases (83%) show improvements over the second-best neural method smaller than 0.1%. Similarly, in [2], 6 out of 12 TSP cases (50%) have similar improvement levels.
>
> [1] Collaboration! Towards Robust Neural Methods for Routing Problems. NeurIPS 2024.
>
> [2] Learning encodings for constructive neural combinatorial optimization needs to regret. AAAI 2024.

---

> > ### Comment · Reviewer_upvg · 2025-04-04
> >
> > When considering sample efficiency, I believe it is more reasonable to focus on the total number of generated samples rather than the amount used for updates. Nevertheless, the other aspects have been addressed well overall. Therefore, I will increase my score by 1 point.

---

> > > ### Author Response · Authors · 2025-04-08
> > >
> > > Thank you for acknowledging our response and for your valuable feedback. We are pleased to address your remain concerns.
> > >
> > > In our experiments, our POCO is fairly compared with state-of-the-art training methods using the same total number of generated samples, demonstrating POCO's higher sample efficiency. Specifically, POCO outperforms the state-of-the-art (self-labeling-based) SLIM on JSP, as shown in Section 5.4 in our submission; POCO surpasses the state-of-the-art (RL-based) POMO and InViT on TSP, as shown in the supplementary experiments in our previous response. Remarkably, in these comparisons, POCO and POMO/InViT both sample $D\times B$ solutions in each batch, but POCO uses only $D\times K$ solutions after uniform filtering for gradient updates, where $D$ is the batch size and $B$ is the number of sampled solutions per instance. This fair comparison further highlights POCO's significantly higher sample efficiency.
> > >
> > > In addition, the full results for 800 epochs on InViT, as mentioned in our previous response, are updated in the table below, consistently demonstrating POCO's higher efficiency and superior performance. The results of InViT are cited from its original paper [1], with $D=128$ and $B=1$. Due to memory constraints, we set $D=64$ and $B=1$ for InViT. For a fair comparison, we set $D=1$ and $B=64$ for InViT+POCO.
> > >
> > > **Table: Gaps (%) Across Different Distributions and Scales**
> > >
> > > |Method|U-1k|U-5k|U-10k|C-1k|C-5k|C-10k|
> > > |-|-|-|-|-|-|-|
> > > |InViT-2V ($D=128,B=1$)|6.15|6.88|6.18|9.32|9.07|9.02|
> > > |InViT-2V ($D=64,B=1$)|6.31|6.94|6.21|9.32|9.07|9.02|
> > > |InViT-2V+POCO ($D=1,B=64$)|**5.13**|**6.54**|**4.97**|**9.13**|**8.96**|**8.68**|
> > >
> > > |Method|E-1k|E-5k|E-10k|I-1k|I-5k|I-10k|
> > > |-|-|-|-|-|-|-|
> > > |InViT-2V ($D=128,B=1$)|9.11|9.92|9.32|6.63|**7.63**|6.78|
> > > |InViT-2V ($D=64,B=1$)|9.26|10.04|9.44|6.69|8.64|6.81|
> > > |InViT-2V+POCO ($D=1,B=64$)|**8.37**|**9.79**|**8.26**|**6.58**|8.57|**5.93**|
> > >
> > > "U": Uniform, "C": Cluster, "E": Explosion, "I": Implosion distributions. The numbers behind represent the problem scales.
> > >
> > > [1] InViT: A generalizable routing problem solver with invariant nested view transformer. ICML 2024.

---

### Decision · Program_Chairs · 2025-05-01

**Decision:**

Accept (poster)

**Comment:**

This paper focuses on neural combinatorial optimization and addresses the issue of sparse rewards. The paper's contributions are to use multiple solutions and provide the information about ordering of the solutions. The empirical evaluation shows clear improvement. All the reviewers viewed the contributions of the paper as significant enough to warrant acceptance.